# A Stronger Benchmark for Online Bilateral Trade:
# From Fixed Prices to Distributions

**Anna Lunghi** [1]  **Mattia Piccinato** [1]  **Matteo Castiglioni** [1]  **Alberto Marchesi** [1]

## Abstract

We study online bilateral trade, where a learner facilitates repeated exchanges between a buyer and a seller to maximize the Gain From Trade (GFT), *i.e.*, the social welfare. In doing so, the learner must guarantee not to subsidize the market. This constraint is usually imposed per round through Weak Budget Balance (WBB). Despite that, Bernasconi et al. (2024) show that a Global Budget Balance (GBB) constraint on the profit—enforced over the entire time horizon—can improve the GFT by a multiplicative factor of two. While this might appear to be a marginal relaxation, this implies that all existing WBB-focused algorithms suffer linear regret when measured against the GBB optimum. In this work, we provide the first algorithm to achieve sublinear regret against the GBB benchmark in stochastic environments under one-bit feedback. In particular, we show that when the joint distribution of valuations has a bounded density, our algorithm achieves $\widetilde{\mathcal{O}}(T^{3/4})$ regret. Our result shows that there is no separation between the one-dimensional problem of learning the optimal WBB price and the two-dimensional problem of learning the optimal GBB distribution over *pairs* of prices.

## 1. Introduction

*Bilateral trade* is a fundamental problem in economics involving a broker and two rational agents: a seller and a buyer. In this setting, agents hold private valuations for a seller's item, and the broker's objective is to design a mechanism that facilitates a transaction between them. Theoretically, the ideal mechanism is ex-post efficient, meaning that it maximizes social welfare by ensuring that a trade occurs whenever the buyer's valuation exceeds the seller's. How-

ever, the landmark result by Myerson & Satterthwaite (1983) establishes that no "economically viable" mechanism can guarantee ex-post efficiency in this context.

To bypass the fundamental limitations of this single-shot setting, a recent line of research (see, *e.g.*, (Cesa-Bianchi et al., 2021; 2024a; Azar et al., 2022; Bernasconi et al., 2024; Lunghi et al., 2025a; Chen et al., 2025)) has shifted focus toward *repeated* bilateral trade. In this framework, a broker interacts with a sequence of buyer-seller pairs over a finite horizon $T$. At each time step $t$, a new pair of agents arrives with private valuations $s_t$ and $b_t$. The broker's task is to select a pair of (potentially randomized) prices $(p_t, q_t)$ to propose to the seller and buyer, respectively. Then, the item is exchanged if and only if both agents find the prices favorable—specifically, when $s_t \leq p_t$ and $b_t \geq q_t$. Upon a successful trade, the broker collects $q_t$ from the buyer and remunerates the seller with $p_t$, gaining $q_t - p_t$ as profit.

The broker's objective is typically to maximize the Gain from Trade (GFT) subject to some profit constraints. These include, in decreasing order of strictness: (i) Strong Budget Balance (SBB), where the broker sets $p_t = q_t$ at every round; (ii) Weak Budget Balance (WBB), in which $p_t \leq q_t$ at every round; and (iii) Global Budget Balance (GBB), where the broker's cumulative profit is non-negative over the entire horizon $T$. A series of papers (Cesa-Bianchi et al., 2021; 2023; Azar et al., 2022) focuses on achieving no regret against the best WBB fixed-price mechanism (or equivalently the best SBB mechanism). This line of research culminated in Bernasconi et al. (2024), who achieved a $\widetilde{\mathcal{O}}(T^{3/4})$ regret bound in adversarial settings—a result proven tight by Chen et al. (2025); Lunghi et al. (2025a).

In this paper, we tackle the more ambitious problem of competing against the best GBB fixed *distribution* over prices. While this might appear to be a marginal relaxation, the GBB benchmark can outperform the best WBB mechanism by a multiplicative factor of two (Bernasconi et al., 2024). Crucially, this implies that all existing WBB-focused algorithms suffer linear regret when evaluated against the GBB optimum. While achieving no regret against a GBB benchmark is known to be impossible in adversarial environments (Bernasconi et al., 2024), the problem remains open for *stochastic* environments—which we address here.

[1]Politecnico di Milano. Correspondence to: Anna Lunghi <anna.lunghi@polimi.it>.

*Proceedings of the 43rd International Conference on Machine Learning*, Seoul, South Korea. PMLR 306, 2026. Copyright 2026 by the author(s).

## 1.1. Our Results and Techniques

At a high level, the main result of this paper is that there is *no separation* between the one-dimensional problem of learning an optimal SBB fixed-price mechanism and the two-dimensional problem of learning an optimal GBB fixed distribution over *pairs* of prices.

We first show that stochasticity alone is insufficient to guarantee sublinear regret. Specifically, we employ a "needle-in-the-haystack" construction to show that, unless the joint distribution of valuations admits a *bounded density*, any algorithm must suffer linear regret. This stands in stark contrast to the SBB setting, where bounded density is unnecessary. The divergence stems from the fact that, for non-budget-balanced price pairs, even a small shift in the prices can cause the expected profit to drop significantly.

Our main result shows that *bounded density is all we need*. Indeed, we design an algorithm that achieves $\widetilde{\mathcal{O}}(T^{3/4})$ regret against an optimal GBB fixed distribution over prices. This rate is tight, as it matches the known $\Omega(T^{3/4})$ lower bound against the more restrictive class of SBB mechanisms, even assuming bounded density (Chen et al., 2025). The main challenges faced by our algorithm are the expanded two-dimensional decision space and the need to handle unknown feasibility constraints. Unlike previous mechanisms that satisfy budget-related constraints deterministically at each round, a GBB mechanism must deal with the feasibility—defined by the non-negativity of aggregate profit—over the whole time horizon.

Our algorithm is organized in three phases. In the *first* one, the learner collets a profit buffer to compensate potential losses incurred during subsequent phases. We prove that the regret incurred during this phase is of the same order as the accumulated profit (up to logarithmic factors). The *second* phase faces one of the main challenges, *i.e.*, the estimation of the GFT. Because GFT does *not* provide bandit-like feedback, it must be learned through pure exploration. This is usually done over a (one-dimensional) grid of size $K$, where it is possible to obtain a uniform $\epsilon$-approximation over the grid with $\widetilde{\mathcal{O}}\left(\frac{K}{\epsilon^2}\right)$ samples. Since we work with GBB mechanisms, we need to estimate the GFT over the two-dimensional grid of $K \times K$ price pairs. This would suggest a significant increase in sample complexity. Surprisingly, we show that it is possible to obtain a uniform approximation over the two-dimensional grid without increasing the number of samples. This is achieved through a clever reuse of samples across different price pairs. Finally, the algorithm enters a *third* phase where it learns the profit function and converges to the optimal distribution over prices. Here, we leverage the stronger bandit feedback available for profit, integrating our previous GFT estimates with an optimism-in-the-face-of-uncertainty principle to identify and exploit an optimal approximately-GBB strategy.

## 1.2. GGB Algorithm Vs. GBB Baseline

The closest to our work is (Bernasconi et al., 2024), which provides a $\widetilde{\mathcal{O}}(T^{3/4})$ regret bound against the best SBB mechanism by using a GBB algorithm. While our approach and results may appear similar at a high level, there is a fundamental distinction between using GBB algorithms and competing against a GBB baseline. Bernasconi et al. (2024) effectively reduce the problem to a single-dimensional one. Indeed, they utilize GBB strategies primarily as a "smoothing" mechanism to bypass the smoothed-adversary assumptions by Cesa-Bianchi et al. (2024b). Once a sufficient profit buffer is collected, their task simplifies to unconstrained regret minimization over a single-dimensional set of $K$ nearly budget-balanced price pairs. In contrast, our setting is inherently two-dimensional. Moreover, rather than restricting the price set *a priori*, we must interact with the environment to learn which distributions over the entire $K \times K$ price grid satisfy budget balance in expectation.

## 1.3. Additional Related Works

Cesa-Bianchi et al. (2024a) initiated the study of online bilateral trade, by showing that, with one-bit feedback, SBB mechanisms are learnable when the seller and buyer valuation distributions are independent and admit bounded density, while the problem becomes unlearnable in general. Subsequent work mainly focuses on adversarial settings. Azar et al. (2022) design a no-2-regret algorithm, while Cesa-Bianchi et al. (2024b) achieve sublinear regret against a smoothed adversary. Bernasconi et al. (2024); Chen et al. (2025) remove the smoothness assumption by considering GBB algorithms, but still mainly focus on competing against an optimal SBB fixed-price mechanism. Lunghi et al. (2026) go even further by analyzing the trade-off between regret and the cumulative violation of GBB constraints.

Other lines of work study extensions of bilateral trade to multiple buyers (Babaioff et al., 2024; Lunghi et al., 2025b), contextual settings (Gaucher et al., 2025), fairer objectives (Bachoc et al., 2024), and symmetric settings with not predetermined agent roles (Bolić et al., 2024; Cesari & Colomboni, 2025; Bachoc et al., 2025a;b). There is also a rich, though less related, literature on offline bilateral trade, which focuses on approximating optimal mechanisms (Colini-Baldeschi et al., 2016; 2017; Blumrosen & Mizrahi, 2016; Brustle et al., 2017; Colini-Baldeschi et al., 2020; Babaioff et al., 2020; Dütting et al., 2021; Deng et al., 2022; Kang & Vondrák, 2019; Archbold et al., 2023).

## 2. Preliminaries

We study repeated bilateral trade in an online learning settings over $T \in \mathbb{N}$ rounds. At each round $t \in [T]$, a new buyer and a new seller arrive, each with a private valuation

of the indivisible item to be exchanged, denoted by $s_t$ and $b_t$, respectively.[1] We focus on a *stochastic* environment, where the private valuations are assumed to be i.i.d. samples from a fixed joint distribution $\mathcal{P}$ with support in $[0, 1]^2$, *i.e.*,

$$(s_1, b_1), (s_2, b_2), \ldots, (s_T, b_T) \overset{i.i.d.}{\sim} \mathcal{P}.$$

Here, we mainly focus on distributions with *bounded density*, namely distributions that admit a density function bounded above by some constant $\sigma \in \mathbb{R}^+$. In the following, we will hide the dependence on $\sigma$ in the $\mathcal{O}(\cdot)$ notation.

**Learning Protocol**   At each round $t \in [T]$, the learner posts two prices: a price $p_t \in [0, 1]$ to the seller and a price $q_t \in [0, 1]$ to the buyer. Notice that the learner does *not* have any knowledge of the private valuations $s_t, b_t$ when posting the prices. Then, the seller accepts the trade if and only if $s_t \leq p_t$, while the buyer does so if and only if $b_t \geq q_t$. Hence, the trade happens whenever both the seller and the buyer accept. As is customary in the literature, the learner's utility is represented by the Gain From Trade (GFT), which corresponds to the difference between the buyer's and the seller's valuations if the trade occurs, and 0 otherwise:

$$\mathsf{GFT}_t(p_t, q_t) := (b_t - s_t) \, \mathbb{I}(s_t \leq p_t) \, \mathbb{I}(b_t \geq q_t).$$

We also define the expected GFT of prices $(p, q) \in [0, 1]^2$ with respect to the joint distribution over valuations as:

$$\mathsf{GFT}(p, q) := \mathbb{E}_{(s,b) \sim \mathcal{P}}[(b - s) \, \mathbb{I}(s \leq p) \, \mathbb{I}(b \geq q)].$$

The GFT represents the increase in social welfare generated by the trade. We will omit the second argument of $\mathsf{GFT}_t$ and $\mathsf{GFT}$ whenever the two arguments are identical.

**Budget Balance**   One of the key concepts in bilateral trade is budget balance. Intuitively, budget balance constraints guarantee that the learner is *not* subsidizing the market. Formally, whenever a trade happens with prices $(p_t, q_t)$, the learner obtains a *profit* equal to $q_t - p_t$. Notice that this quantity can be either positive or negative. Formally,

$$\mathsf{PRO}_t(p_t, q_t) := (q_t - p_t) \, \mathbb{I}(s_t \leq p_t) \, \mathbb{I}(b_t \geq q_t).$$

If the prices are such that $\mathsf{PRO}_t(p_t, q_t) = 0$ for all $t \in [T]$ (which happens when $p_t = q_t$ for all $t$), the mechanism is called *Strongly* Budget Balanced (SBB). If the prices satisfy $\mathsf{PRO}_t(p_t, q_t) \geq 0$ for all $t \in [T]$ (which happens when $p_t \leq q_t$ for all $t$), the mechanism is called *Weakly* Budget Balanced (WBB). Moreover, if the mechanism satisfies

$$\sum_{t=1}^{T} \mathsf{PRO}_t(p_t, q_t) \geq 0,$$

---

[1] In this paper, we let $[k]$ with $k \in \mathbb{N}$ be the set of all the natural numbers from 1 to $k$, *i.e.*, $[k] := \{1, 2, \ldots, k\}$.

it is said to be *Globally* Budget Balanced (GBB). Unlike the previous notions, GBB does *not* impose constraints at each individual round, but only cumulatively over all rounds, thus allowing for more flexibility. Similarly to the GFT, we define the expected profit of prices $(p, q) \in [0, 1]^2$ with respect to the joint distribution over valuations as

$$\mathsf{PRO}(p, q) := \mathbb{E}_{(s,b) \sim \mathcal{P}}[(q - p) \, \mathbb{I}(s \leq p) \, \mathbb{I}(b \geq q)].$$

**Feedback Model**   Traditionally, three types of feedback are considered in online bilateral trade:

- *Full feedback*: the learner directly observes the private valuations $(s_t, b_t)$ at the end of each round $t \in [T]$.

- *Two-bit feedback*: the learner observes the decisions of the seller and the buyer separately, *i.e.*, the values of $\mathbb{I}(s_t \leq p_t)$ and $\mathbb{I}(b_t \geq q_t)$.

- *One-bit feedback*: the learner only observes whether the trade happens or *not*, *i.e.*, $\mathbb{I}(s_t \leq p_t)\mathbb{I}(b_t \geq q_t)$.

**Regret Against the Best GBB Fixed Distribution**   The last component required to define our setting is a benchmark to measure the performance of learning algorithms. The first works on the topic take as a benchmark a fixed-price mechanism. Formally, the fixed price $(p, p)$ that maximizes the expected GFT. Notice that setting $p = q$, *i.e.*, posting the same price to the buyer and the seller, is without loss of generality even when optimizing over WBB mechanisms. However, this benchmark does *not* exploit the flexibility that GBB provides. For this reason, Bernasconi et al. (2024) introduce a stronger baseline requiring that the profit is non-negative *in expectation*. This is natural and more aligned with the line of research on constrained problems and bandits with knapsack (see, *e.g.*, (Badanidiyuru et al., 2018; Immorlica et al., 2022; Castiglioni et al., 2022)). Formally, we take as baseline the *best fixed distribution over prices that is budget balanced in expectation*. Let $\Delta([0, 1]^2)$ be the family of all probability measures over the space $([0, 1]^2, \mathcal{B}([0, 1]^2))$, where $\mathcal{B}$ denotes the Borel $\sigma$-algebra. Then, we define:

$$\mathsf{OPT} := \max_{\gamma \in \Delta([0,1]^2)} \mathbb{E}_{(p,q) \sim \gamma}[\mathsf{GFT}(p, q)] \quad \text{s.t.}$$

$$\mathbb{E}_{(p,q) \sim \gamma}[\mathsf{PRO}(p, q)] \geq 0,$$

so that the *regret* over the time horizon $T$ is defined as

$$R_T := T \cdot \mathsf{OPT} - \sum_{t=1}^{T} \mathsf{GFT}(p_t, q_t).$$

**Problem Formulation Recap**   In this paper, we focus on the problem of minimizing the regret $R_T$, *i.e.*, the regret with respect to the best distribution that is budget balanced in expectation, while at the same time guaranteeing that the learning algorithm is GBB. Moreover, we assume the more stringent form of feedback, *i.e.*, one-bit feedback.

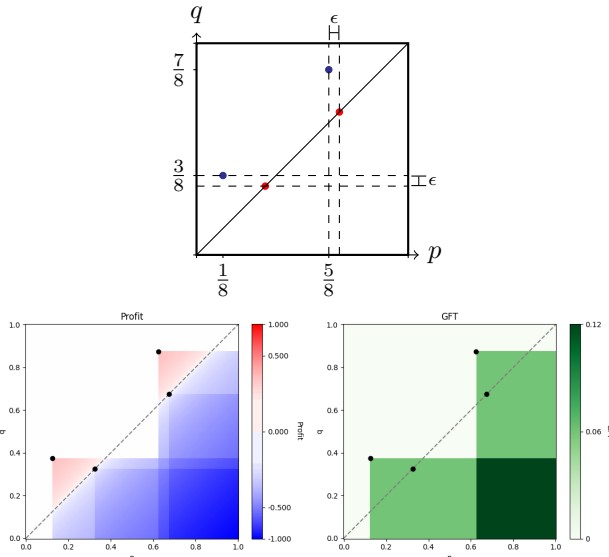

*Figure 1. Top*: Support of the probability distribution $\mathcal{D}_\epsilon$. *Bottom*: Expected profit (*left*) and GFT (*right*) under $\mathcal{D}_\epsilon$.

## 3. Why We Need Bounded Density

In this section, we show that, without the bounded density assumption, it is impossible to achieve sublinear regret, even under two-bit feedback. As usual, this is done by constructing a family of hard instances for which achieving sublinear regret for any instance is impossible. We use the "needle in the haystack" argument to show that the crucial obstacle to learning in the general setting is the highly non-continuous nature of the GFT and PRO functions, which make any form of discretization uneffective. This observation motivates the assumption of bounded density, which we maintain for the remaining of the paper.

### 3.1. Construction of the Hard Instances

We parametrize an hard instance by a parameter $\epsilon$. In particular, we let $\mathcal{D}_\epsilon$ be a probability distribution in the space $([0,1]^2, \mathcal{B}([0,1]^2))$, characterized by the probability function $\mathbb{P}_{\mathcal{D}_\epsilon} : \mathcal{B}([0,1]^2) \to [0,1]$, such that

$$\begin{cases} \mathbb{P}_{\mathcal{D}_\epsilon}((1/8, 3/8)) = 1/4 \\ \mathbb{P}_{\mathcal{D}_\epsilon}((5/8, 7/8)) = 1/4 \\ \mathbb{P}_{\mathcal{D}_\epsilon}((3/8 - \epsilon, 3/8 - \epsilon)) = 1/4 \\ \mathbb{P}_{\mathcal{D}_\epsilon}((5/8 - \epsilon, 5/8 - \epsilon)) = 1/4, \end{cases}$$

It is easy to observe that any algorithm failing to identify the region $(3/8 - \epsilon, 3/8] \times [5/8, 5/8 + \epsilon)$ must incur in linear regret. Indeed, this either results in a loss of GFT or increases the probability of trades with zero GFT, which in turn reduces the expected profit. This must be compensated by price pairs with large profit and small GFT, incurring into a loss in GFT. Since $\epsilon$ can be arbitrarily small, even

slightly perturbing the instance (by slightly perturbing $3/8$ and $5/8$) makes identifying these regions infeasible by standard information-theoretic arguments. Figure 1 provides a graphical depiction of the hard instances.

**Theorem 3.1.** *For any $T \in \mathbb{N}$ and any learning algorithm, there is an instance such that $\mathbb{E}[R_T] \geq \Omega(T)$.*

## 4. Main Theorem and Proof Plan

The main contribution of this paper is to show that, under the assumption that the distribution over valuations admits bounded density, it is possible to extend the $\widetilde{\mathcal{O}}(T^{3/4})$ regret bound of Bernasconi et al. (2024) to a stronger baseline: *the best GBB fixed distribution over prices*. Formally:

**Theorem 4.1.** *Assume that $\mathcal{P}$ admits a density bounded by $\sigma \in \mathbb{R}_+$ and let $\delta \in (0,1)$. Then, with probability at least $1 - \delta$, Algorithm 1 is GBB and guarantees*

$$R_T \leq \widetilde{\mathcal{O}}(T^{3/4}).$$

The proof of the theorem is structured as follows. In Section 5, we show that the bounded density assumption enables the discretization of the problem onto a finite grid—a reduction that is impossible otherwise, as discussed in Section 3. In Section 6, we provide a high-level overview of the algorithm that achieves no regret relative to an optimal distribution over price pairs in this grid. Section 7, Section 8, and Section 9 then detail and analyze the three phases of the algorithm. Finally, Section 10 exploits these results to complete the proof of Theorem 4.1.

## 5. Under Bounded Density a Grid is Enough!

In this section, we discuss how assuming that $\mathcal{P}$ has bounded density allows us to circumvent the linear lower bound on the regret. This is achieved by restricting to a finite grid. Specifically, we show that for any feasible distribution $\gamma$ over $([0,1]^2, \mathcal{B}([0,1]^2))$—that is, any distribution with non-negative expected profit—there exists a distribution $\gamma_K$ supported on a uniform grid of $K^2$ points such that

$$\mathbb{E}_{(p,q)\sim\gamma}[\mathsf{GFT}(p,q)] - \mathbb{E}_{(p,q)\sim\gamma_K}[\mathsf{GFT}(p,q)] \leq \mathcal{O}\left(\frac{1}{K}\right)$$

$$\mathbb{E}_{(p,q)\sim\gamma_K}[\mathsf{PRO}(p,q)] \geq 0.$$

Let $\mathcal{G}_K$ be the uniform grid

$$\mathcal{G}_K := \left\{ \left( \frac{i}{K-1}, \frac{j}{K-1} \right) : i,j \in \{0, \ldots, K-1\} \right\}.$$

We first show that, under the bounded density assumption, "projecting" a feasible distribution onto $\mathcal{G}_K$ changes the expected GFT by at most $\mathcal{O}(1/K)$, and that the obtained distribution is unfeasible by at most a factor $\mathcal{O}(1/K)$.

**Lemma 5.1.** *Let $\gamma \in \Delta([0,1]^2)$ be a feasible distribution with density function $f_\gamma$. Moreover, let $\widehat{\gamma}_K$ be a distribution over $\mathcal{G}_K$ defined as follows: for all $(p,q) \in \mathcal{G}_K$*

$$\widehat{\gamma}_K(p,q) := \mathbb{P}_{(x,y)\sim\gamma}\left((x,y) \in (p - \tfrac{1}{K}, p] \times [q, q + \tfrac{1}{K})\right).$$

*Then, the following conditions hold:*

$$\mathbb{E}_{(p,q)\sim\gamma}[\mathsf{GFT}(p,q)] - \mathbb{E}_{(p,q)\sim\widehat{\gamma}_K}[\mathsf{GFT}(p,q)] \leq \frac{2\sigma}{K-1}$$

*and*

$$\mathbb{E}_{(p,q)\sim\widehat{\gamma}_K}[\mathsf{PRO}(p,q)] \geq -\frac{2\sigma}{K-1}.$$

Next, we show that, starting from the "projected" distribution, one can build another distribution that is (exactly) feasible, by only losing a factor $\mathcal{O}(\log(T)/K)$ in terms of expected GFT. This allows us to prove that there exists a feasible distribution over $\mathcal{G}_K$ that achieves an optimal expected GFT up to an additive error $\mathcal{O}(\log(T)/K)$. Formally, let $\mathsf{OPT}_K$ denote the value of an optimal distribution over prices that is budget balanced in expectation, *i.e.*,

$$\mathsf{OPT}_K := \max_{\gamma\in\Delta(\mathcal{G}_K)} \mathbb{E}_{(p,q)\sim\gamma}[\mathsf{GFT}(p,q)] \quad \text{s.t.} \tag{1a}$$

$$\mathbb{E}_{(p,q)\sim\gamma}[\mathsf{PRO}(p,q)] \geq 0. \tag{1b}$$

It is then possible to prove the following lemma.

**Lemma 5.2.** *For any $K \in \mathbb{N}$, it holds*

$$\mathsf{OPT} - \mathsf{OPT}_K \leq \mathcal{O}\left(\frac{\log(T)}{K}\right).$$

*Proof Sketch.* Let $\gamma^\star \in \Delta([0,1]^2)$ be an optimal feasible distribution, so that

$$\mathsf{OPT} = \mathbb{E}_{(p,q)\sim\gamma^\star}[\mathsf{GFT}(p,q)].$$

Let $\widehat{\gamma}_K^*$ denote its "projection" onto $\mathcal{G}_K$. Let $(p^\star, q^\star) \in \mathcal{G}_K$ be a price pair maximizing the expected profit, and define

$$\gamma_K^\star := (1-\alpha)\widehat{\gamma}_K^* + \alpha\delta_{(p^\star,q^\star)},$$

for a suitable choice of $\alpha$, where $\delta_{(p^\star,q^\star)}$ is the probability distribution that puts all probability mass in $(p^\star, q^\star)$. Then:

$$\begin{aligned}
\mathsf{OPT} - \mathsf{OPT}_K &\leq (1-\alpha)(\mathsf{OPT} - \mathbb{E}_{(p,q)\sim\widehat{\gamma}_K^*}[\mathsf{GFT}]) \\
&\quad + \alpha(\mathsf{OPT} - \mathsf{GFT}(p^\star, q^\star)) \\
&\leq \frac{2\sigma}{K-1} + \alpha\mathsf{OPT},
\end{aligned}$$

where the second inequality follows from Lemma 5.1.

Intuitively, if $\mathsf{PRO}(p^\star, q^\star)$ is large, feasibility can be restored with a very small $\alpha$. If instead $\mathsf{PRO}(p^\star, q^\star)$ is small,

the connection between GFT and PRO implies that OPT itself must be sublinear. By using this relation, it holds:

$$\alpha\mathsf{OPT} \leq \mathcal{O}\left(\frac{\log(T)}{K}\right),$$

which leads to the result. We defer the formal proof of this lemma to Section B in the appendix. $\square$

## 6. High-Level Construction of the Algorithm

As shown in the previous section, the bounded density assumption allows us to restrict the attention to a uniform grid $\mathcal{G}_K$ of $K^2$ points. This results into a cumulative discretization error of $\mathcal{O}\left(\frac{T}{K}\right)$ for both GFT and profit. This error term characterizes the trade-off between the samples needed to explore the price space (which increase with $K$) and the discretization error (which decreases with $K$). Once the optimal value of $K$ is picked, it suffices to design an algorithm that operates only on the discretized price space.

---

**Algorithm 1**

---

**Require:** $T \in \mathbb{N}, \delta \in (0,1)$
 1: Set $\beta, \mathcal{F}_K, \mathcal{G}_K, N$ as functions of $T$ and $\delta$
 2: Run $\mathsf{Profit\text{-}Max}(\mathcal{F}_K, \beta, T)$
 3: $(\overline{R}, \overline{L}) \leftarrow \mathsf{Simultaneous\text{-}Exploration}(\mathcal{G}_K, N)$
 4: Run $\mathsf{Explore\text{-}Exploit}(\mathcal{G}_K\overline{R}, \overline{L}, N, T, \delta)$

---

Our algorithm is conceptually divided into three parts (see Algorithm 1), each with its own specific objective:

- **Profit collection phase.** The first phase is dedicated to collecting profit. Here, we use a profit maximization algorithm, called $\mathsf{Profit\text{-}Max}$, until a target profit $\beta$ is reached. This allows us to explore safely in the subsequent phases, even by (slightly) violating the budget constraint. This revenue maximization phase was introduced by Bernasconi et al. (2024) together with the notion of GBB. Interestingly, they show that collecting budget does *not* lead to catastrophic regret, as one might expect. In particular, the collected profit is almost linear in the GFT of the optimal fixed-price mechanism. In this phase, our main contribution is to show that a similar result holds when comparing against the best distribution over price pairs.

- **Pure exploration phase.** The second phase consists of pure exploration, and it is performed by the algorithm $\mathsf{Simultaneous\text{-}Exploration}$ over the grid $\mathcal{G}_K$. Indeed, it is well known that, for GFT, one-bit feedback is much weaker than bandit feedback. In particular, picking a pair of prices $(p,q)$ does *not* suffice to obtain any information about the expected GFT of $(p,q)$. Estimating the GFT may require exploring other pairs of prices

that can potentially be arbitrarily worse. To address this exploration challenge, we decompose $\mathsf{GFT}(p, q)$ into two components: one that is not directly observable by selecting $(p, q)$ and must be estimated via a pure exploration strategy, and one that comes with bandit feedback, allowing exploration and exploitation to be performed simultaneously by choosing $(p, q)$. In this phase, the algorithm learns the "pure exploration" component, while the "bandit" component is learned in the next phase. The primary challenge here lies in the two-dimensional nature of our price set, which distinguishes our setting from prior work. While standard methods only require learning the GFT for SBB mechanisms (*i.e.*, $K$ prices), we need estimates for the entire grid of $K^2$ prices, including prices that are *not* budget balanced. Our key contribution in this phase is to prove that these $K^2$ estimates can be obtained by exploring only $2K$ price pairs. This shows that expanding the estimation phase from SBB to arbitrary prices does *not* increase the required sample complexity.

- **GFT optimization phase.** The third phase consists of the actual GFT optimization, which is performed by Explore-Exploit. In this phase, we use the estimates obtained during exploration to construct an equivalent problem that can be formulated as a constrained online optimization problem with bandit feedback. Specifically, we use the estimate of the non-directly observable component of the GFT as a bias applied to the bandit loss corresponding to the directly observable component. This step is fundamental: by using a suitably adapted optimistic approach, we obtain a dependence on the number of actions of $\mathcal{O}(\sqrt{K^2})$ rather than the $\mathcal{O}(K^2)$ dependence that would arise from a pure explore-then-commit approach.

## 7. Profit Collection

Algorithm 1 starts by collecting profit. Indeed, given the global budget balance constraint, the idea is to accumulate some budget in order to gain the flexibility needed to explore the space of possible price distributions in later phases.

The general intuition behind this phase is the following. If the instance is "simple" enough, meaning that there is an easy way to accumulate positive profit, this phase will do so rapidly. If the instance is instead "hard", and the possibility of accumulating profit is limited, then profit maximization will take longer (possibly lasting for the entirety of the rounds), but the regret incurred will also be small.

In practice, the algorithm Profit-Max instantiates a regret minimizer over the additive-multiplicative grid $\mathcal{F}_K$ introduced by Bernasconi et al. (2024). For the definition of $\mathcal{F}_K$ and a more details, see Section C in the appendix.

**Lemma 7.1.** *Given the additive-multiplicative grid $\mathcal{F}_K$ by Bernasconi et al. (2024), it holds that*

$$\mathsf{OPT} \leq 16 \log T \sup_{(p,q) \in \mathcal{F}_K} \mathsf{PRO}(p, q) + \frac{10}{K}.$$

*Proof Sketch.* We start relating $\mathsf{OPT}$ to the GFT of the best fixed price, with a multiplicative factor of 2. By applying Theorem 6.3 in (Bernasconi et al., 2024), we obtain:

$$\mathsf{OPT} \leq 2 \sup_{p \in [0,1]} \mathsf{GFT}(p, p).$$

Then, by restricting to a one-dimensional set of prices (*i.e.*, $p = q$), it is possible to construct an additive-multiplicative grid similar to the one proposed by Bernasconi et al. (2024). This grid has cardinality $\mathcal{O}(\log(T)K)$ and guarantees that

$$\max_{p \in [0,1]} \mathsf{GFT}(p, p) \leq 8 \log(T) \max_{(p,q) \in \mathcal{F}_K} \mathsf{PRO}(p, q) + \frac{5}{K},$$

as desired. $\qquad\square$

Implementing a regret minimizer over $\mathcal{F}_K$ that maximizes profit until a threshold is reached (see Algorithm 4 in Section C in the appendix) yields the following lemma.

**Lemma 7.2.** *With proability at least $1 - \delta$, if Profit-Max with target budget $\beta$ stops at round $\tau \in [T]$, then*

$$\tau\mathsf{OPT} - \sum_{t=1}^{\tau}\mathsf{GFT}(p_t, q_t) \leq \widetilde{\mathcal{O}}\left(\beta + \frac{T}{K} + \sqrt{KT \log(1/\delta)}\right).$$

## 8. Compressing the Exploration Time

A fundamental challenge of learning in online bilateral trade is the inherent tension between exploration and exploitation. Under one-bit or two-bit feedback, the GFT of a chosen price pair $(p, q)$ is never directly observable. This limitation has been leveraged in prior work to establish lower bounds on learning rates when buyer and seller valuations are correlated. Specifically, bilateral trade can be viewed as a variant of "apple-tasting", where determining the optimal pair of prices requires choosing prices that are clearly suboptimal. Consequently, in this setting, even optimal learning algorithms require a dedicated exploration phase. Our objective is to optimize this phase, showing how state-of-the-art approaches for SBB mechanisms can be extended to price pairs without changing the sample complexity.

The general idea is to decompose the GFT into two main components: one that corresponds to an integral term, *i.e.*, the part of the GFT that requires an expensive pure exploration phase, and one that can be estimated while exploiting (akin to bandit feedback). To this end, we start from the

following decomposition of the GFT:

$$\mathsf{GFT}(p,q) = \int_0^p \mathbb{P}(s \leq x, q \leq b)\, dx$$

$$+ \int_q^1 \mathbb{P}(s \leq p, x \leq b)\, dx + (q-p)\,\mathbb{P}(s \leq p, q \leq b).$$

**Lemma 8.1.** *Let $U, V$ be independent random variables distributed uniformly over $[0,1]$. Then, the expected GFT generated by prices $(p,q) \in [0,1]^2$ can be decomposed as*

$$\mathsf{GFT}(p,q) = \mathbb{E}_{U,(s,b)\sim\mathcal{P}}[\mathbb{P}(s \leq U \leq p, q \leq b)]$$
$$+ \mathbb{E}_{V,(s,b)\sim\mathcal{P}}[\mathbb{P}(s \leq p, q \leq V \leq b)] + \mathsf{PRO}(p,q).$$

The lemma is a direct consequence of the integral decomposition and the properties of the uniform distribution. For all $(p,q) \in [0,1]^2$, we introduce $L(p,q)$ and $R(p,q)$ such that:

$$L(p,q) \coloneqq \mathbb{E}_{U,(s,b)\sim\mathcal{P}}[\mathbb{P}(s \leq U \leq p, q \leq b)]$$
$$R(p,q) \coloneqq \mathbb{E}_{V,(s,b)\sim\mathcal{P}}[\mathbb{P}(s \leq p, q \leq V \leq b)].$$

This decomposition is particularly useful, as it separates the bandit-feedback component, namely the profit term, from the non-directly observable components $L$ and $R$. The goal of the pure exploration phase is therefore to efficiently learn estimates of these two quantities for each $(p,q) \in \mathcal{G}_K$.

The decomposition above allows us to reduce the original task of estimating $K^2$ quantities—one for each pair of prices in the grid $\mathcal{G}_K$—to estimating only $2K$ quantities of the form $\mathbb{P}(s \leq U, q \leq b)$ and $\mathbb{P}(s \leq p, V \leq b)$. These estimates can be combined with the profit term to build an estimate of the GFT for all the $K^2$ price pairs. This reduction from $K^2$ to $2K$ is a crucial step, as the regret depends linearly on this quantity. This is achieved by observing that

$$\mathbb{I}(s \leq U \leq p, q \leq b) = \mathbb{I}(s \leq U, q \leq b)\mathbb{I}(U \leq p).$$

This implies that an unbiased estimator $\widehat{L}$ can be built simultaneously for all $p$ by choosing a pair of prices $(U, q)$, with $U$ sampled uniformly from $[0,1]$, and multiplying the observed feedback bit $\mathbb{I}(s \leq U)\mathbb{I}(q \leq b)$ by the indicator $\mathbb{I}(U \leq p)$, which can be computed for all $p$ since $U$ is known. Naturally, an analogous argument applies to constructing an unbiased estimator $\widehat{R}$. Finally, by applying Hoeffding's inequality, we obtain the following lemma.

**Lemma 8.2.** *Let $\delta \in (0,1)$. If Algorithm 2 terminates before round $T$, then, with probability at least $1 - \delta$:*

$$\left|\widehat{L}(p,q) - L(p,q)\right| \leq \sqrt{\frac{\log\left(\frac{4K^2}{\delta}\right)}{N}}$$

$$\left|\widehat{R}(p,q) - R(p,q)\right| \leq \sqrt{\frac{\log\left(\frac{4K^2}{\delta}\right)}{N}}$$

*for all $(p,q) \in \mathcal{G}_K$, and it runs for $2KN$ rounds.*

---

**Algorithm 2** Simultaneous-Exploration

---

**Require:** Grid $\mathcal{G}_K$, $N \in \mathbb{N}$
1: **for all** $i \in \{0, \ldots, K-1\}$ **do**
2:     $q \leftarrow \frac{i}{K-1}, \tau \leftarrow t + N$
3:     **while** $t \leq \tau$ **do**
4:        Sample $U_t$ uniformly in $[0,1]$, post prices $(U_t, q)$ and observe $\mathbb{I}(s_t \leq U_t, q \leq b_t)$
5:        **for all** $p : (p,q) \in \mathcal{G}_K$ **do**
6:           $\widehat{L}(p,q) \leftarrow \widehat{L}(p,q) + \frac{1}{N}\mathbb{I}(s_t \leq U_t, q \leq b_t)\mathbb{I}(U_t \leq p)$
7:        **end for**
8:        $t \leftarrow t + 1$
9:     **end while**
10: **end for**
11: **for all** $i \in \{0, \ldots, K-1\}$ **do**
12:     $p \leftarrow \frac{i}{K-1}, \tau \leftarrow t + N$
13:     **while** $t \leq \tau$ **do**
14:        Sample $V_t$ uniformly in $[0,1]$, post prices $(p, V_t)$ and observe $\mathbb{I}(s_t \leq p, q \leq V_t)$
15:        **for all** $q : (p,q) \in \mathcal{G}_K$ **do**
16:           $\widehat{R}(p,q) \leftarrow \widehat{R}(p,q) + \frac{1}{N}\mathbb{I}(s_t \leq p_t, b \geq V_t)\mathbb{I}(V_t \geq q)$
17:        **end for**
18:        $t \leftarrow t + 1$
19:     **end while**
20: **end for**
21: **return** functions $\widehat{L}(\cdot, \cdot)$ and $\widehat{R}(\cdot, \cdot)$

---

Notice that it is impossible to compute a good estimation of the profit over the grid $\mathcal{G}_K$, as it would require to select $N$ times all the $K^2$ pairs of prices. In contrast, our characterization allows us to use a single sample to compute and estimation of the GFT of $K$ differently couples. This reduces the number of samples per pair to $N/K$, greatly reducing the length of the pure exploration phase.

## 9. Constrained Bandit Optimization

In the previous section, we isolated the components of the GFT that do *not* provide bandit feedback, and we showed how it is possible to learn those components uniformly and efficiently. We now focus on the profit term. This term has both advantages and drawbacks. Indeed, an explore-and-commit framework is ineffective in this setting, since the profit of each price must be learned independently. However, the presence of bandit feedback allows for simultaneous exploration and exploitation. We exploit this property by using a UCB-like approach. We face an additional challenge that we have ignored for most of the paper. Indeed, unlike all previous works, we must work with distributions subject to *unknown constraints*. We handle these constraints by using an optimistic estimation. Here, our profit-collection phase plays a crucial role, by compensating for the loss in profit due to the optimistic constraints.

Formally, Algorithm 1 reformulates Problem 1 as

$$\max_{\gamma \in \Delta(\mathcal{G}_K)} \mathbb{E}_{(p,q)\sim\gamma}[L(p,q) + R(p,q) + \mathsf{PRO}(p,q)] \quad \text{s.t.}$$

$$\mathbb{E}_{(p,q)\sim\gamma}[\mathsf{PRO}(p,q)] \geq 0,$$

where the $L$ and $R$ components can be interpreted as action-specific biases for which we already have accurate estimates, while $\mathsf{PRO}(p,q)$ represents the utility component equipped with bandit feedback. Thus, we can tackle this problem as a standard bandit-feedback constrained optimization problem, adjusted through bias terms, and solve it by using an optimistic approach. In particular, at each interaction, the algorithm updates its optimistic estimates of the profit as

$$\overline{\mathsf{PRO}}_t(p,q) = \frac{1}{N_t(p,q)} \sum_{\tau=\tau_0}^{t} \mathsf{PRO}_t(p,q)\mathbb{I}(p_\tau = p, q_\tau = q)$$

$$+ \min\left\{1, \sqrt{\frac{2\log\left(\frac{6TK^2}{\delta}\right)}{N_t(p,q)}}\right\}, \tag{2}$$

where $\tau_0$ is the round in which the algorithm is initialized and $N_t(p,q)$ is defined as the number of times in which the algorithm has chosen the prices $(p,q)$ up to round $t$.

In Algorithm 3, we provide the complete pseudocode of the procedure. The following lemma provides its guarantees.

**Lemma 9.1.** *Let $\gamma \in \Delta(\mathcal{G}_K)$ be a distribution over $\mathcal{G}_K$ such that $\mathbb{E}_{(p,q)\sim\gamma}[\mathsf{PRO}(p,q)] \geq 0$. If Algorithm 3 is initialized at round $\tau_0 \in [T]$, with probability at least $1 - \delta$:*

$$\sum_{t=\tau_0}^{T}\left(\sum_{(p,q)\in\mathcal{G}_K} \gamma(p,q)r(p,q) - r(p_t,q_t)\right)$$

$$\leq \widetilde{\mathcal{O}}\left(K\sqrt{T\log(1/\delta)}\right),$$

$$\sum_{t=\tau_0}^{T} \mathsf{PRO}(p_t,q_t) \geq -\widetilde{\mathcal{O}}\left(K\sqrt{T\log(1/\delta)}\right),$$

*where we let $r(p,q) := \overline{R}(p,q) + \overline{L}(p,q) + \mathsf{PRO}(p,q)$.*

## 10. Putting Everything Together

Now, we are finally ready to combine the results on the three phases to prove Theorem 4.1. In the following, we provide a proof sketch and postpone the proof to Section F. To bound the regret, we decompose it into five components:

- The regret incurred by restricting attention to the discretized price set $\mathcal{G}_K$, which is of order $T/K$.

- The regret $\widetilde{\mathcal{O}}(\beta + \sqrt{KT} + \frac{T}{K})$ in the profit collection phase, of order , where $\beta$ denotes the budget accumulated. We set $\beta = \widetilde{\mathcal{O}}(NK + \sqrt{K^2T\ln(1/\delta)})$ to

---

**Algorithm 3** Explore-Exploit

**Require:** $\mathcal{G}_K$, functions $\widehat{L}$ and $\widehat{R}$, $N, T \in \mathbb{N}$, $\delta \in (0,1)$
1: **for all** $(p,q) \in \mathcal{G}_K$ **do**
2: $\quad \overline{R}(p,q) \leftarrow \widehat{R}(p,q) + \sqrt{\frac{\log(\frac{4K^2}{\delta})}{N}}$
3: $\quad \overline{L}(p,q) \leftarrow \widehat{L}(p,q) + \sqrt{\frac{\log(\frac{4K^2}{\delta})}{N}}$
4: $\quad \overline{\mathsf{PRO}}_t(p,q) \leftarrow 1$
5: $\quad \gamma_t(p,q) \leftarrow \frac{1}{|\mathcal{G}_K|}$
6: **end for**
7: $\tau_0 \leftarrow t$
8: **while** $t \leq T$ **do**
9: $\quad$ Sample $(p_t,q_t) \sim \gamma_t$ and observe $\mathsf{PRO}_t(p_t,q_t)$
10: $\quad$ Update estimates $\overline{\mathsf{PRO}}_t(\cdot,\cdot)$ as in Equation (2)
11: $\quad$ Update optimistic biased reward for all $(p,q) \in \mathcal{G}_K$:

$$r_t(p,q) \leftarrow \overline{R}(p,q) + \overline{L}(p,q) + \overline{\mathsf{PRO}}_t(p,q)$$

12: $\quad$ Compute:

$$\gamma_{t+1} \leftarrow \underset{\gamma\in\Delta(\mathcal{G}_K)}{\arg\max} \sum_{(p,q)\in\mathcal{G}_K} r_t(p,q)\gamma(p,q) \quad \text{s.t.}$$

$$\sum_{(p,q)\in\mathcal{G}_K} \overline{\mathsf{PRO}}_t(p,q)\gamma(p,q) \geq 0$$

13: $\quad t \leftarrow t + 1$
14: **end while**

---

account for both the profit loss during the pure exploration phase and the losses incurred by defining the space of feasible distributions through optimism.

- The regret of the pure exploration phase, which is bounded by the number of its rounds, namely $2KN$.

- The regret in the constrained bandit optimization phase, generated by using an optimistic estimate of $L$ and $R$ rather than their real values, of the order $\mathcal{O}(\frac{T}{\sqrt{N}})$.

- The regret incurred during the constrained bandit optimization phase with respect to the reward functions defined through the optimistic estimates of $L$ and $R$, which is proportional to the square root of the number of actions multiplied by the square root of the number of rounds, i.e., $\widetilde{\mathcal{O}}(\sqrt{K^2T\ln(1/\delta)})$.

Combining these terms, the total regret satisfies

$$R_T \leq \widetilde{\mathcal{O}}\left(T/K + NK + \sqrt{K^2T\ln(1/\delta)}\right) = \widetilde{\mathcal{O}}\left(T^{3/4}\right),$$

with $N = \Theta(T^{1/2})$ and $K = \Theta(T^{1/4})$. The optimistic approach guarantees that this bound holds with high probability; more precisely, the above bound holds with probability at least $1 - \delta$, as stated in the theorem.

## Impact Statement

This paper presents work whose goal is to advance the field of Machine Learning. There are many potential societal consequences of our work, none which we feel must be specifically highlighted here.

## Acknowledgments

This paper is supported by the Italian MIUR PRIN 2022 Project "Targeted Learning Dynamics: Computing Efficient and Fair Equilibria through No-Regret Algorithms", by the FAIR (Future Artificial Intelligence Research) project, funded by the NextGenerationEU program within the PNRRPE-AI scheme (M4C2, Investment 1.3, Line on Artificial Intelligence), and by the EU Horizon project ELIAS (European Lighthouse of AI for Sustainability, No. 101120237)

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

# A. Omitted Proofs from Section 3

**Theorem 3.1.** *For any $T \in \mathbb{N}$ and any learning algorithm, there is an instance such that $\mathbb{E}[R_T] \geq \Omega(T)$.*

*Proof.* We construct a family of instances parameterized by $\epsilon > 0$ and $u \in [-1/16, 1/16]$. Each instance is characterized by a probability measure

$$\mathbb{P}_{\mathcal{D}_{\epsilon,u}} : \mathcal{B}([0,1]^2) \rightarrow [0,1],$$

defined by

$$\mathbb{P}_{\mathcal{D}_{\epsilon,u}}\big((1/8,\, 3/8 + u)\big) = \tfrac{1}{4},$$
$$\mathbb{P}_{\mathcal{D}_{\epsilon,u}}\big((5/8 + u,\, 7/8)\big) = \tfrac{1}{4},$$
$$\mathbb{P}_{\mathcal{D}_{\epsilon,u}}\big((3/8 + u - \epsilon,\, 3/8 + u - \epsilon)\big) = \tfrac{1}{4},$$
$$\mathbb{P}_{\mathcal{D}_{\epsilon,u}}\big((5/8 + u - \epsilon,\, 5/8 + u - \epsilon)\big) = \tfrac{1}{4}.$$

We partition the $(p, q)$-space into regions and characterize, for each region, the resulting GFT and profit.

- **Region I:**
$$(p, q) \in (3/8 + u - \epsilon,\, 3/8 + u] \times [5/8 + u,\, 5/8 + u + \epsilon).$$

   In this region, trades corresponding to the positive-GFT valuation pairs $(1/8, 3/8 + u)$ and $(5/8 + u, 7/8)$ are realized, while trades with zero GFT are excluded. Therefore,
$$\mathsf{GFT}(p, q) = \tfrac{2}{16},$$
$$\mathsf{PRO}(p, q) \geq -\tfrac{2}{8} \cdot \tfrac{1}{2}.$$

- **Region II:**
$$(p, q) \in [0,\, 5/8 + v) \times (3/8 + u,\, 1].$$

   In this region, no trade is executed. Hence,
$$\mathsf{GFT}(p, q) = 0,$$
$$\mathsf{PRO}(p, q) = 0.$$

- **Region III:**
$$(p, q) \in [5/8 + u,\, 1] \times [0,\, 3/8 + u] \setminus (3/8 + u - \epsilon,\, 3/8 + u] \times [5/8 + u,\, 5/8 + u + \epsilon).$$

   Here, all positive-GFT trades are realized, but at least some zero-GFT trades are also executed, resulting in a loss in profit. In particular,
$$\mathsf{GFT}(p, q) = \tfrac{2}{16},$$
$$\mathsf{PRO}(p, q) \leq -\tfrac{2}{8} \cdot \tfrac{3}{4}.$$

- **Region IV:** For any other choice of $(p, q)$, we have
$$\mathsf{GFT}(p, q) \leq \tfrac{1}{16} + \frac{u}{4},$$
$$\mathsf{PRO}(p, q) \leq \left(\tfrac{2}{8} + |u|\right) \cdot \tfrac{1}{4}.$$

As a consequence, there exists a constant $c > 0$ such that the optimal policy—mixing between Region I and Region IV achieves a GFT that exceeds by at least $c$ that of any feasible policy that never selects prices from Region I.

This follows from the observation that playing with constant probability outside Region I either directly reduces the GFT or induces negative profit. Any such loss in profit must then be compensated by increasing the probability of selecting prices from Region IV that are linearly suboptimal in terms of GFT, leading to a net loss bounded away from zero.

Finally, by standard information-theoretic arguments, identifying an interval of length $\epsilon$ within an interval of length $1/8$ requires at least $\Omega(\log(1/\epsilon))$ rounds in the worst case. Given any time horizon $T$, taking $\epsilon$ sufficiently small, the algorithm is unable to reliably identify Region I with high probability, and consequently incurs linear regret:

$$\mathbb{E}[R_T] \geq \Omega(T).$$

$\square$

# B. Omitted Proofs from Section 5

**Lemma 5.1.** *Let $\gamma \in \Delta([0,1]^2)$ be a feasible distribution with density function $f_\gamma$. Moreover, let $\widehat{\gamma}_K$ be a distribution over $\mathcal{G}_K$ defined as follows: for all $(p,q) \in \mathcal{G}_K$*

$$\widehat{\gamma}_K(p,q) := \mathbb{P}_{(x,y)\sim\gamma}\left((x,y) \in (p - \frac{1}{K}, p] \times [q, q + \frac{1}{K})\right).$$

*Then, the following conditions hold:*

$$\mathbb{E}_{(p,q)\sim\gamma}[\mathsf{GFT}(p,q)] - \mathbb{E}_{(p,q)\sim\widehat{\gamma}_K}[\mathsf{GFT}(p,q)] \leq \frac{2\sigma}{K-1}$$

*and*

$$\mathbb{E}_{(p,q)\sim\widehat{\gamma}_K}[\mathsf{PRO}(p,q)] \geq -\frac{2\sigma}{K-1}.$$

*Proof.* Let $(p,q) \in [0,1]^2$, and let $(\Pi_K^s(p), \Pi_K^b(q))$ denote the nearest point in the grid $\mathcal{G}_K$ such that $\Pi_K^s(p) \geq p$ and $\Pi_K^b(q) \leq q$.

We decompose the gain-from-trade as follows:

$$
\begin{aligned}
GFT(p,q) &= \mathbb{E}\big[(b-s)\mathbb{I}(s \leq p,\, b \geq q)\big] \\
&= \mathbb{E}\big[(b-s)\mathbb{I}(s \leq \Pi_K^s(p),\, b \geq \Pi_K^b(q))\big] \\
&\quad - \mathbb{E}\big[(b-s)\mathbb{I}(p < s \leq \Pi_K^s(p),\, b < q)\big] \\
&\quad - \mathbb{E}\big[(b-s)\mathbb{I}(s \leq p,\, \Pi_K^b(q) \leq b < q)\big] \\
&\quad + \mathbb{E}\big[(b-s)\mathbb{I}(p \leq s < \Pi_K^s(p),\, \Pi_K^b(q) < b \leq q)\big].
\end{aligned}
$$

Using $|b - s| \leq 1$ and bounded density, we obtain

$$
\begin{aligned}
GFT(p,q) &\leq \mathbb{E}\big[(b-s)\mathbb{I}(s \leq \Pi_K^s(p),\, b \geq \Pi_K^b(q))\big] \\
&\quad + \mathbb{P}(p \leq s < \Pi_K^s(p),\, q \leq b) + \mathbb{P}(s < p,\, \Pi_K^b(q) < b \leq q) \\
&= GFT(\Pi_K^s(p), \Pi_K^b(q)) + \frac{2\sigma}{K}.
\end{aligned}
$$

Given any feasible distribution $\gamma \in \Delta([0,1]^2)$, define the discretized distribution $\hat{\gamma}_K$ by

$$\hat{\gamma}_K(p,q) = \mathbb{P}_{(x,y)\sim\gamma}\left((x,y) \in (p - \frac{1}{K}, p] \times [q, q + \frac{1}{K})\right) \quad \forall(p,q) \in \mathcal{G}_K$$

Then,

$$
\begin{aligned}
\mathbb{E}_{(p,q)\sim\gamma}[GFT(p,q)] &\leq \mathbb{E}_{(p,q)\sim\gamma}\left[GFT(\Pi_K^s(p), \Pi_K^b(q)) + \frac{2\sigma}{K}\right] \\
&\leq \mathbb{E}_{(p,q)\sim\hat{\gamma}_K}[GFT(p,q)] + \frac{2\sigma}{K}.
\end{aligned}
$$

A similar argument yields

$$\mathsf{PRO}(p,q) \leq \mathsf{PRO}(\Pi_K^s(p), \Pi_K^b(q)) + \mathbb{P}(p \leq s < \Pi_K^s(p), \, q \leq b)$$
$$+ \mathbb{P}(s < p, \, \Pi_K^b(q) < b \leq q)$$
$$\leq \mathsf{PRO}(\Pi_K^s(p), \Pi_K^b(q)) + \frac{2\sigma}{K},$$

and therefore

$$\mathbb{E}_{(p,q)\sim\gamma}[\mathsf{PRO}(p,q)] \leq \mathbb{E}_{(p,q)\sim\hat{\gamma}_K}[\mathsf{PRO}(p,q)] + \frac{2\sigma}{K}.$$

By feasibility of $\gamma$,

$$\mathbb{E}_{(p,q)\sim\hat{\gamma}_K}[\mathsf{PRO}(p,q)] \geq -\frac{2\sigma}{K}.$$

$\square$

**Lemma 5.2.** *For any $K \in \mathbb{N}$, it holds*

$$\mathsf{OPT} - \mathsf{OPT}_K \leq \mathcal{O}\left(\frac{\log(T)}{K}\right).$$

*Proof.* Let

$$(p^+, q^+) \in \arg\max_{(p,q)\in\mathcal{G}_K} \mathsf{PRO}(p,q),$$

and define

$$\gamma_K = (1-\alpha)\hat{\gamma}_K + \alpha\delta_{(p^+,q^+)}, \quad \alpha := \frac{2\sigma/K}{\mathsf{PRO}(p^+,q^+)},$$

where $\delta_{(p^+,q^+)}$ is a distribution that assign probability 1 to the couple of prices $(p^+, q^+)$.

Therefore, we can study two separate cases, $\mathsf{PRO}(p^+,q^+) \geq \frac{2\sigma}{K}$ and $\mathsf{PRO}(p^+,q^+) < \frac{2\sigma}{K}$.

**Case 1: $\mathsf{PRO}(p^+,q^+) \geq \frac{2\sigma}{K}$.** If $\mathsf{PRO}(p^+,q^+) \geq \frac{2\sigma}{K}$, then $\alpha$ belongs to $[0,1]$ and $\gamma_K$ is well defined. In addition, by construction,

$$\mathbb{E}_{(p,q)\sim\gamma_K}[\mathsf{PRO}(p,q)] = (1-\alpha)\mathbb{E}_{(p,q)\sim\hat{\gamma}_K}[\mathsf{PRO}(p,q)] + \alpha\mathsf{PRO}(p^+,q^+)$$
$$\geq (1-\alpha)\frac{-2\sigma}{K} + \frac{2\sigma}{K} \geq 0.$$

and we can bound the expected gain-from-trade using the followings :

- Using Lemma 7.1 and bounded density, one has

$$\mathsf{OPT} \leq 16\log T \cdot \max_{(p,q)\in\mathcal{F}_K} \mathsf{PRO}(p,q) + \frac{10}{K}$$
$$\leq 16\log T \cdot \max_{(p,q)\in\mathcal{G}_K} \mathsf{PRO}(p,q) + 16\log T \cdot \frac{2\sigma}{K} + \frac{10}{K}$$
$$= 16\log T \cdot \mathsf{PRO}(p^+,q^+) + 16\log T \cdot \frac{2\sigma}{K} + \frac{10}{K}$$

- By definition of $\alpha$,

$$\alpha\mathsf{PRO}(p^+,q^+) = \frac{2\sigma}{K}.$$

Therefore,

$$OPT - OPT_K \leq (1-\alpha)\left(\frac{-2\sigma}{K}\right) + \alpha\big(OPT - \mathsf{GFT}(p^+,q^+)\big)$$
$$\leq \frac{-2\sigma}{K} + \alpha OPT$$
$$\leq \mathcal{O}\left(\log(T)\frac{1}{K}\right).$$

**Case 2: $\mathsf{PRO}(p^+,q^+) < \frac{2\sigma}{K}$.** Finally, to conclude the proof, we study the case in which $\mathsf{PRO}(p^+,q^+) < \frac{2\sigma}{K}$, and therefore $\gamma_K$ is not well defined.

In this case, however, we observe that also the optimum must be small, indeed

$$\mathsf{OPT} \leq 16\log T \cdot \mathsf{PRO}(p^+,q^+) + 16\log T \cdot \frac{2\sigma}{K} + \frac{10}{K}$$
$$\leq 16\log T \cdot \frac{2\sigma}{K} + +16\log T \cdot \frac{2\sigma}{K} + \frac{10}{K}.$$

Therefore, to conclude the proof it is sufficient to observe that

$$OPT - OPT_K \leq OPT \leq \mathcal{O}\left(\log(T)\frac{1}{K}\right),$$

since $\mathsf{OPT}_K \geq 0$, as choosing any price on the diagonal $(p = q)$ cannot yeld negative GFT. $\qquad\square$

# C. Omitted Proofs from Section 7

In this section we present the omitted details from Section 7 .

As explained in the main paper, the profit maximization phase is almost identical to the one introduced by (Bernasconi et al., 2024). Their construction is based on defining a $\mathcal{O}(\log_2(T)K)$ dimensional grid, over which we can instantiate a regret minimizer algorithm with the goal of maximizing the Profit. Indeed, the profit benefits from bandit feedback, making maximizing the profit on a grid a Multi-armed Bandit problem.

First, we define the grid $\mathcal{F}_K$ as the union of two different grids $\mathcal{F}_K^-$ and $\mathcal{F}_K^+$, i.e. :

$$\mathcal{F}_K^- := \left\{(q - 2^{-i}, q) : q \text{ s.t. } (q,q) \in \mathcal{G}_K \text{ and } i \in \{0,1,\dots,\lceil\log(T)\rceil\}\right\} \cap [0,1]^2$$
$$\mathcal{F}_K^+ := \left\{(p, p + 2^{-i}) : p \text{ s.t. } (p,p) \in \mathcal{G}_K \text{ and } i \in \{0,1,\dots,\lceil\log(T)\rceil\}\right\} \cap [0,1]^2$$
$$\mathcal{F}_K := \mathcal{F}_K^+ \cup \mathcal{F}_K^-$$

This type of multiplicative grid was first introduced by (Bernasconi et al., 2024). With a similar analysis to theirs we can prove the following lemma, which correlates the optimum $GFT$ achievable by a feasible distribution on $\mathcal{G}_K$ with the optimal revenue obtainable on the grid $\mathcal{F}_K$ .

**Lemma 7.1.** *Given the additive-multiplicative grid $\mathcal{F}_K$ by Bernasconi et al. (2024), it holds that*

$$\mathsf{OPT} \leq 16\log T \sup_{(p,q)\in\mathcal{F}_K} \mathsf{PRO}(p,q) + \frac{10}{K}.$$

*Proof.* Let $p^* \in [0,1]$ be defined as

$$p^* = \arg\max_{p\in[0,1]} GFT(p,p).$$

Then, the proof come directly by combining Theorem 6.3 of (Bernasconi et al., 2024),

$$OPT \leq 2\max_{p\in[0,1]} GFT(p,p),$$

and Lemma 5.1 of (Bernasconi et al., 2024).

$\qquad\square$

---

**Algorithm 4** Revenue Maximization [(Bernasconi et al., 2024)]

---

1: Require grid $\mathcal{F}_K$, target budget $\beta$, number of rounds $T$
2: Initialize regret minimizer $\mathcal{A}$ as Exp3
3: $B \leftarrow 0$
4: **while** $t \leq T, B < \beta$ **do**
5:    Choose $(p_t, q_t) \in \mathcal{F}_K$ suggested by $\mathcal{A}$
6:    Update $\mathcal{A}$ with $\mathsf{PRO}_t(p_t, q_t)$
7:    $B \leftarrow B + \mathsf{PRO}_t(p_t, q_t)$
8:    $t \leftarrow t + 1$
9: **end while**

---

## D. Omitted Proof from Section 8

**Lemma 8.1.** *Let $U, V$ be independent random variables distributed uniformly over $[0, 1]$. Then, the expected GFT generated by prices $(p, q) \in [0, 1]^2$ can be decomposed as*

$$\mathsf{GFT}(p, q) = \mathbb{E}_{U,(s,b)\sim\mathcal{P}}[\mathbb{P}(s \leq U \leq p, q \leq b)]$$
$$+ \mathbb{E}_{V,(s,b)\sim\mathcal{P}}[\mathbb{P}(s \leq p, q \leq V \leq b)] + \mathsf{PRO}(p, q).$$

*Proof.* By the law of total expectation

$$\mathbb{E}_{U,(s,b)\sim\mathcal{P}}[\mathbb{P}(s \leq U \leq p, q \leq b)] = \mathbb{E}_{(s,b)\sim\mathcal{P}}\left[\mathbb{E}_U[\mathbb{P}(s \leq U \leq p, q \leq b)|s, b]\right]$$
$$= \mathbb{E}_{(s,b)\sim\mathcal{P}}[(p - s)\mathbb{I}(s \leq p, q \leq b)]$$

and similarly

$$\mathbb{E}_{V,(s,b)\sim\mathcal{P}}[\mathbb{P}(s \leq p, q \leq V \leq b)] = \mathbb{E}_{(s,b)\sim\mathcal{P}}\left[\mathbb{E}_V[\mathbb{P}(s \leq p, q \leq V \leq b)|s, b]\right]$$
$$= \mathbb{E}_{(s,b)\sim\mathcal{P}}[(b - q)\mathbb{I}(s \leq p, q \leq b)]$$

therefore

$$\mathbb{E}_{U,(s,b)\sim\mathcal{P}}[\mathbb{P}(s \leq U \leq p, q \leq b)] + \mathbb{E}_{V,(s,b)\sim\mathcal{P}}[\mathbb{P}(s \leq p, q \leq V \leq b)] + \mathsf{PRO}(p, q) =$$
$$= \mathbb{E}_{(s,b)\sim\mathcal{P}}[(p - s)\mathbb{I}(s \leq p, q \leq b)] + \mathbb{E}_{(s,b)\sim\mathcal{P}}[(b - q)\mathbb{I}(s \leq p, q \leq b)] + \mathbb{E}_{(s,b)\sim\mathcal{P}}[(q - p)\mathbb{I}(s \leq p, q \leq b)]$$
$$= \mathbb{E}_{(s,b)\sim\mathcal{P}}[((b - q) + (p - s) + (q - p))\,\mathbb{I}(s \leq p, q \leq b)]$$
$$= \mathbb{E}_{(s,b)\sim\mathcal{P}}[(b - s)\mathbb{I}(s \leq p, q \leq b)] = \mathsf{GFT}(p, q).$$

$\square$

**Lemma 8.2.** *Let $\delta \in (0, 1)$. If Algorithm 2 terminates before round T, then, with probability at least $1 - \delta$:*

$$\left|\widehat{L}(p, q) - L(p, q)\right| \leq \sqrt{\frac{\log\left(\frac{4K^2}{\delta}\right)}{N}}$$

$$\left|\widehat{R}(p, q) - R(p, q)\right| \leq \sqrt{\frac{\log\left(\frac{4K^2}{\delta}\right)}{N}}$$

*for all $(p, q) \in \mathcal{G}_K$, and it runs for $2KN$ rounds.*

*Proof.* By construction both $\widehat{L}(p, q)$ and $\widehat{R}(p, q)$ are the empirical mean of $N$ unbiased estimators of $L(p, q)$ and $R(p, q)$ respectively, for all $(p, q) \in \mathcal{G}_K$. Hence, by Hoeffding inequality, fixed a $(p, q)$, with probability at least $1 - \delta'$

$$\left|\widehat{L}(p, q) - L(p, q)\right| \leq \sqrt{\frac{\log(\frac{2}{\delta'})}{N}},$$

and similarly, fixed a $(p, q)$, with probability at least $1 - \delta'$

$$\left| \widehat{R}(p, q) - R(p, q) \right| \leq \sqrt{\frac{\log(\frac{2}{\delta'})}{N}}.$$

To conclude the proof it is sufficient to take the union bound over all events defined above for all prices $(p, q) \in \mathcal{G}_\mathcal{K}$, and set $\delta = 2K^2 \delta'$. $\qquad\square$

# E. Omitted Proof from Section 9

**Lemma 9.1.** *Let $\gamma \in \Delta(\mathcal{G}_K)$ be a distribution over $\mathcal{G}_K$ such that $\mathbb{E}_{(p,q)\sim\gamma}[\mathsf{PRO}(p, q)] \geq 0$. If Algorithm 3 is initialized at round $\tau_0 \in [T]$, with probability at least $1 - \delta$:*

$$\sum_{t=\tau_0}^{T} \left( \sum_{(p,q)\in\mathcal{G}_K} \gamma(p, q) r(p, q) - r(p_t, q_t) \right)$$

$$\leq \widetilde{\mathcal{O}}\left( K\sqrt{T \log(1/\delta)} \right),$$

$$\sum_{t=\tau_0}^{T} \mathsf{PRO}(p_t, q_t) \geq -\widetilde{\mathcal{O}}\left( K\sqrt{T \log(1/\delta)} \right),$$

*where we let $r(p, q) := \overline{R}(p, q) + \overline{L}(p, q) + \mathsf{PRO}(p, q)$.*

*Proof.* Let $\gamma \in \Delta(\mathcal{G}_K)$ be an arbitrary distribution over the grid $\mathcal{G}_K$ such that

$$\sum_{(p,q)\in\mathcal{G}_K} \gamma(p, q) \mathsf{PRO}(p, q) \geq 0.$$

We define the event $\mathcal{E}$ as

$$\left| \frac{1}{N_t(p, q)} \sum_{\tau=\tau_0}^{T} \mathsf{PRO}(p, q)\mathbb{I}(p = p_\tau, q = q_\tau) - \mathsf{PRO}(p, q) \right| \leq \phi_t(p, q) \quad \forall (p, q) \in \mathcal{G}_K, \, \forall t \in \{\tau_0, \dots, T\}.$$

Since $\mathsf{PRO}(p, q) \in [-1, 1]$, by Hoeffding's inequality and a union bound over all rounds $t$ and all pairs of prices $(p, q) \in \mathcal{G}_K$, it holds that $\mathcal{E}$ occurs with probability at least $1 - \delta/4$, with

$$\phi_t(p, q) = \min\left\{ 1, \sqrt{\frac{2}{N_t(p, q)} \log\left( \frac{8TK^2}{\delta} \right)} \right\}.$$

In particular, by definition

$$\overline{\mathsf{PRO}}_t(p, q) = \frac{1}{N_t(p, q)} \sum_{\tau=\tau_0}^{T} \mathsf{PRO}(p, q)\mathbb{I}(p = p_\tau, q = q_\tau) + \phi_t(p, q).$$

Therefore, under the event $\mathcal{E}$, the distribution $\gamma$ belongs to the feasible region of the linear program of Line 13,

$$\sum_{(p,q)\in\mathcal{G}_K} \gamma(p, q) \overline{\mathsf{PRO}}_t(p, q) \geq \sum_{(p,q)\in\mathcal{G}_K} \gamma(p, q) \mathsf{PRO}(p, q) \geq 0.$$

In addition, using the symmetry of the confidence bounds and the optimality of $\gamma_{t+1}$ for the linear program, we obtain

$$\sum_{(p,q)\in\mathcal{G}_K} \gamma_{t+1}(p,q)r(p,q) \geq \sum_{(p,q)\in\mathcal{G}_K} \gamma_{t+1}(p,q)\big(\overline{R}(p,q) + \overline{L}(p,q)\big)$$
$$+ \sum_{(p,q)\in\mathcal{G}_K} \gamma_{t+1}(p,q)\big(\overline{\mathsf{PRO}}_t(p,q) - 2\phi_t(p,q)\big)$$
$$\geq \sum_{(p,q)\in\mathcal{G}_K} \gamma_{t+1}(p,q)r_t(p,q) - 2\sum_{(p,q)\in\mathcal{G}_K} \gamma_{t+1}(p,q)\phi_t(p,q)$$
$$\geq \sum_{(p,q)\in\mathcal{G}_K} \gamma(p,q)r_t(p,q) - 2\sum_{(p,q)\in\mathcal{G}_K} \gamma_{t+1}(p,q)\phi_t(p,q)$$
$$\geq \sum_{(p,q)\in\mathcal{G}_K} \gamma(p,q)r(p,q) - 2\sum_{(p,q)\in\mathcal{G}_K} \gamma_{t+1}(p,q)\phi_t(p,q).$$

Summing over $t = \tau_0, \ldots, T$, we obtain

$$\sum_{t=\tau_0}^{T} \sum_{(p,q)\in\mathcal{G}_K} \gamma_{t+1}(p,q)r(p,q) \geq \sum_{t=\tau_0}^{T} \sum_{(p,q)\in\mathcal{G}_K} \gamma(p,q)r(p,q) - 2\sum_{t=\tau_0}^{T} \sum_{(p,q)\in\mathcal{G}_K} \gamma_{t+1}(p,q)\phi_t(p,q).$$

By applying the Azuma–Hoeffding inequality, with probability at least $1 - \delta/4$,

$$\sum_{t=\tau_0}^{T} \sum_{(p,q)\in\mathcal{G}_K} \gamma_{t+1}(p,q)\phi_t(p,q) \leq \sum_{t=\tau_0}^{T} \phi_t(p_{t+1}, q_{t+1}) + \sqrt{2\log(8/\delta)|T - \tau_0 + 1|}$$
$$\leq \sqrt{2\log\left(\tfrac{8T}{\delta}\right)} \sum_{t=\tau_0}^{T} \frac{1}{\sqrt{N_t(p_{t+1}, q_{t+1})}} + \sqrt{2\log(8/\delta)|T - \tau_0 + 1|}$$
$$= \sqrt{2\log\left(\tfrac{8T}{\delta}\right)} \sum_{t=\tau_0}^{T} \sum_{(p,q)\in\mathcal{G}_K} \frac{\mathbb{I}\big((p,q) = (p_{t+1}, q_{t+1})\big)}{\sqrt{N_t(p,q)}}$$
$$+ \sqrt{2\log(8/\delta)|T - \tau_0 + 1|}$$
$$\leq 2\sqrt{2\log\left(\tfrac{8T}{\delta}\right)|\mathcal{G}_K||T - \tau_0 + 1|} + \sqrt{2\log(8/\delta)|T - \tau_0 + 1|}$$

By applying the Azuma-Hoeffding inequality once more, considering that $r(p,q) \leq 3$ for any $(p,q)$, with probability at least $1 - \delta/4$,

$$\sum_{t=\tau_0}^{T} \sum_{(p,q)\in\mathcal{G}_K} \gamma_{t+1}(p,q)r(p,q) \leq \sum_{t=\tau_0}^{T} r(p_{t+1}, q_{t+1}) + 3\sqrt{2\log(8/\delta)|T - \tau_0 + 1|}.$$

Finally, again, by applying the Azuma-Hoeffding inequality with probability at least $1 - \delta/4$,

$$\sum_{t=\tau_0}^{T} \sum_{(p,q)\in\mathcal{G}_K} \gamma_t(p,q)\mathsf{PRO}(p,q) \leq \sum_{t=\tau_0}^{T} \mathsf{PRO}(p_t, q_t) + \sqrt{2\log(8/\delta)|T - \tau_0 + 1|}.$$

In conclusion, joining everything, with probability at least $1 - \delta$,

$$\sum_{t=\tau_0}^{T} \sum_{(p,q)\in\mathcal{G}_K} \gamma(p,q)r(p,q) - \sum_{t=\tau_0}^{T} r(p_t, q_t) \leq \widetilde{\mathcal{O}}\Big(K\sqrt{\log(1/\delta)(T - \tau_0 + 1)}\Big),$$

and

$$\sum_{t=\tau_0}^{T} \mathsf{PRO}(p_t, q_t) \geq \sum_{t=\tau_0}^{T} \sum_{(p,q)\in\mathcal{G}_K} \gamma_t(p,q)\mathsf{PRO}(p,q) - \sqrt{2\log(8/\delta)|T-\tau_0+1|}$$

$$\geq -1 + \sum_{t=\tau_0+1}^{T} \sum_{(p,q)\in\mathcal{G}_K} \gamma_t(p,q)\overline{\mathsf{PRO}}_{t-1}(p,q) - 2\sum_{t=\tau_0}^{T} \sum_{(p,q)\in\mathcal{G}_K} \gamma_{t+1}(p,q)\phi_t(p,q)$$

$$- \sqrt{2\log(8/\delta)|T-\tau_0+1|}$$

$$\geq -1 - 4\sqrt{2\log\left(\tfrac{8T}{\delta}\right)K^2|T-\tau_0+1|} - 2\sqrt{2\log(8/\delta)|T-\tau_0+1|}$$

$$- \sqrt{2\log\left(8/\delta\right)|T-\tau_0+1|}$$

$$\geq -\widetilde{\mathcal{O}}(K\sqrt{T\log(1/\delta)}).$$

$\square$

# F. Proof of Theorem 4.1

**Theorem 4.1.** *Assume that $\mathcal{P}$ admits a density bounded by $\sigma \in \mathbb{R}_+$ and let $\delta \in (0,1)$. Then, with probability at least $1-\delta$, Algorithm 1 is GBB and guarantees*

$$R_T \leq \widetilde{\mathcal{O}}(T^{3/4}).$$

*Proof.* Let $\tau_1$ and $\tau_2$, with $\tau_1 \leq \tau_2 \leq T$, be the time in which the phase Profit-Max ends and in which the phase Simultaneous-Exploration ends respectively.

Assume the non-trivial case $\tau_2 < T$. Then, the regret can be decomposed in the following way

$$R_T = \sum_{t=1}^{\tau_1}(OPT - GFT(p_t, q_t)) + \sum_{t=\tau_1+1}^{\tau_2}(OPT - GFT(p_t, q_t)) + \sum_{t=\tau_2+1}^{T}(OPT - GFT(p_t, q_t)).$$

By Lemma 7.2 we can bound the first component as

$$\sum_{t=1}^{\tau_1}(OPT - GFT(p_t, q_t)) \leq \mathcal{O}\left(\beta + \frac{T}{K} + \sqrt{KT}\right)$$

The second component can be bounded using the number of rounds, by Lemma 8.2,

$$\sum_{t=\tau_1+1}^{\tau_2}(OPT - GFT(p_t, q_t)) \leq |\tau_2 - \tau_1| \leq 2NK.$$

We can therefore focus on the third component.

Let $\gamma_K^*$ the distribution over the grid $\mathcal{G}_K$ that achieve expected GFT $\mathsf{OPT}_K$ and expected Profit greater or equal than $0$, as defined in Lemma 5.2. Then, we can further decompose the third component in 4 components:

$$\sum_{t=\tau_2+1}^{T} (OPT - GFT(p_t, q_t)) = \sum_{t=\tau_2+1}^{T} (OPT - OPT_K)$$

$$+ \sum_{t=\tau_2+1}^{T} \left( \mathbb{E}_{(p,q)\sim\gamma_K^*}[L(p,q) + R(p,q)] - \mathbb{E}_{(p,q)\sim\gamma_K^*}[\overline{L}(p,q) + \overline{R}(p,q)] \right)$$

$$+ \sum_{t=\tau_2+1}^{T} \left( \mathbb{E}_{(p,q)\sim\gamma_K^*}[r(p,q)] - r(p_t, q_t) \right)$$

$$+ \sum_{t=\tau_2+1}^{T} \left( (\overline{L}(p_t, q_t) + \overline{R}(p_t, q_t)) - (L(p_t, q_t) + R(p_t, q_t)) \right)$$

We can focus on each of the four components separately:

1. **First component**. It represents the approximation error generated by an algorithm that restricts the space of its action to the grid $\mathcal{G}_K$. By Lemma 5.2

$$\sum_{t=\tau_2+1}^{T} (OPT - OPT_K) \le (T - \tau_2)(\mathsf{OPT} - \mathsf{OPT}_K) \le \widetilde{\mathcal{O}}\left(\frac{T}{K}\right).$$

2. **Second component**. The second component is generated by considering the optimistic estimate of $L$ and $R$ instead of their real value to define the optimum in this phase. However, this , with probability at least $1 - \delta$ , by Lemma 8.2 , is such that

$$\sum_{t=\tau_2+1}^{T} \left( \mathbb{E}_{(p,q)\sim\gamma_K^*}[L(p,q) + R(p,q)] - \mathbb{E}_{(p,q)\sim\gamma_K^*}[\overline{L}(p,q) + \overline{R}(p,q)] \right) \le 0$$

3. **Third component**. The third component is generated by employing UCB like algorithm to maximize the reward function $r$. By Lemma 9.1

$$\sum_{t=\tau_2+1}^{T} \left( \mathbb{E}_{(p,q)\sim\gamma_K^*}[r(p,q)] - r(p_t, q_t) \right) \le \widetilde{\mathcal{O}}\left(K\sqrt{T\log(1/\delta)}\right)$$

4. **Fourth component**. Similarly to the second component it is generated by employing an optimistic estimate of $L$ and $R$ instead to their true value to build the reward function to maximize. By Lemma 8.2

$$\sum_{t=\tau_2+1}^{T} \left( (\overline{L}(p_t, q_t) + \overline{R}(p_t, q_t)) - (L(p_t, q_t) + R(p_t, q_t)) \right) \le \widetilde{\mathcal{O}}\left(\frac{T\log(1/\delta)}{\sqrt{N}}\right).$$

Hence, setting $K = T^{1/4}$, $N = T^{1/2}$ and $\beta = \widetilde{\Theta}(T^{3/4})$, we get that with probability at least $1 - \delta$

$$R_T \le \mathcal{O}(T^{3/4}).$$

To conclude, we prove that the algorithm is Global budget balance, i.e $\sum_{t\in[T]} \mathsf{PRO}(p_t, q_t) \ge 0$. Indeed, under the same clean event defined by Lemma 9.1, with probability at least $1 - \delta$

$$\sum_{t\in[T]} \mathsf{PRO}(p_t, q_t) = \sum_{t=1}^{\tau_1} \mathsf{PRO}(p_t, q_t) + \sum_{t=\tau_1+1}^{\tau_2} \mathsf{PRO}(p_t, q_t) + \sum_{t=\tau_2+1}^{T} \mathsf{PRO}(p_t, q_t)$$

$$\ge \beta - |\tau_2 - \tau_1| - \widetilde{\mathcal{O}}\left(K\sqrt{T\log(1/\delta)}\right) \ge 0,$$

thanks to the fact that the first phase stops only if $\sum_{t=1}^{\tau_1} \mathsf{PRO}(p_t, q_t) \ge \beta$, by Lemma 9.1 $\sum_{t=\tau_2+1}^{T} \mathsf{PRO}(p_t, q_t) \le \widetilde{\mathcal{O}}(K\sqrt{T\log(1/\delta)})$, and for a choice of $\beta = \widetilde{\Theta}\left(NK + K\sqrt{T\log(1/\delta)} + \frac{T}{K}\right) = \widetilde{\Theta}(T^{3/4})$.

$\square$

