# OpenReview forum: "A Stronger Benchmark for Online Bilateral Trade: From Fixed Prices to Distributions"
_ICML.cc/2026/Conference — ICML 2026 regular_

### Official Review · Reviewer_5YL2 · 2026-02-24

**Soundness:** 3
**Presentation:** 2
**Significance:** 2
**Originality:** 3
**Overall Recommendation:** 4
**Confidence:** 3

**Summary:**

This paper proposes upgrading the benchmark in online bilateral trade from fixed (WBB/SBB) prices to the stronger GBB fixed distribution benchmark, under stochastic i.i.d. valuations and one-bit feedback. It proves that without a bounded density assumption, sublinear regret is impossible (via a needle-in-the-haystack construction), and then shows bounded density suffices to discretize to a grid with small loss. The main contribution is presenting the first algorithm that is globally budget balanced (GBB) and achieves $\tilde{O}(T^{3/4})$ regret against the optimal GBB distribution, matching the known $\Omega(T^{3/4})$ barrier already present for SBB baselines, thereby arguing “no separation” between the 1D SBB learning problem and the 2D GBB-distribution learning problem.

**Compliance With Llm Reviewing Policy:**

Affirmed.

**Final Justification:**

Overall, I think the paper is okay, but I still have reservations about the feasibility of some of the assumptions in the paper.

**Key Questions For Authors:**

1. Your main positive theorem uses bounded density. Can you identify weaker regularity assumptions that still support discretization and sublinear regret against the GBB benchmark?
2. The algorithm depends on several horizon-linked parameters, including $K$, $N$, and $\beta$. Is there a clean anytime variant, or can you clarify what specifically breaks when $T$ is unknown?
3. Beyond i.i.d. stochastic valuations, do the ideas extend to mildly non-stationary or contextual settings, and if not, which component becomes the main bottleneck: discretization, exploration, or constrained optimization over feasible distributions?

**Limitations:**

No. The paper would benefit from a short discussion of how the i.i.d. and bounded-density assumptions limit applicability in practice, and whether deploying GBB trading policies could create unintended market effects beyond the theoretical model.

**Strengths And Weaknesses:**

**Strengths**

I think the strongest contribution here is conceptual: the paper changes the benchmark, and that substantially changes what the online learner is actually expected to compete against. The impossibility-plus-constructive-result structure is also well laid out. On top of that, the exploration-compression idea is technically appealing.
- The benchmark upgrade addresses a real issue. Competing with the optimal GBB price distribution is strictly stronger than competing with WBB or SBB fixed prices, and earlier algorithms can indeed incur linear regret under this comparator.
- The paper cleanly distinguishes what is impossible without bounded density from what becomes achievable once bounded density is assumed, which makes the role of the assumption explicit rather than buried.
- The exploration-compression step is clever: it reuses samples to estimate grid components with only $2K$ explored families instead of a naive $K^2$ blow-up.
- Achieving regret of order $T^{3/4}$ up to logarithmic terms while matching existing lower bounds suggests the rate is essentially optimal in this regime.

**Weaknesses**

My main concerns are about accessibility and robustness of assumptions. Bounded density is doing a lot of work, and the algorithm seems fairly phase-heavy and parameterized. Even for a theory paper, I think a light numerical demonstration would make the benchmark shift easier to digest.
- The bounded-density assumption is central, and I am not sure how stable the discretization argument is under weaker regularity, such as nearly bounded densities, piecewise densities, or mild dependence between buyer and seller values.
- The algorithm appears horizon-dependent and involves several tuning parameters. Clearer guidance on parameter choice, or at least a discussion of how the method behaves when $T$ is unknown, would improve usability.
- The paper is almost entirely theoretical. A small synthetic experiment could help readers see when the GBB distribution benchmark materially changes outcomes relative to simpler WBB or SBB baselines.

---

> ### Author Rebuttal · Authors · 2026-03-31
>
> We thank the Reviewer for their comments and for the time devoted to reviewing our paper.
>
> > My main concerns are about accessibility and robustness of assumptions. Bounded density is doing a lot of work, and the algorithm seems fairly phase-heavy and parameterized. Even for a theory paper, I think a light numerical demonstration would make the benchmark shift easier to digest.
>
> We thank the Reviewer for the suggestion, we will add a numerical example to show how the two benchmarks may differ, possibly with a graphical illustration to help conveying the shift. We provide a simple numerical example below.
>
> **Numerical example**
>
> Consider valuations $(s,b)∈\{(0,1/4),(3/4,1)\}$ with probability $1/2$ each.
>
> Any *fixed-price* bechmark can make the trade happen for only one of the two valuations pairs above. Thus, it achieves $GFT = 1/2 \times 1/4 = 0.125$.
>
> Consider instead a *distributional* benchmark that randomizes between the following two price pairs:
> - $(p,q)=(3/4,1/4)$ with probability $1/5$, and
> - $(p,q)=(0,1/4)$ with probability $4/5$.
>
> This results in a profit of $1/5 \times (−1/2)+4/5\times (1/8)=0$, and, thus, it is feasible. Moreover, it achieves $GFT = 1/5 \times (1/4)+4/5 \times (1/8)=0.150$.
>
> Hence, the distributional benchmark is strictly better ($0.150>0.125$). It succeeds by using surplus from "cheap" trades to cross-subsidize "expensive" trades that a single fixed price cannot execute without violating individual rationality or budget balance.
>
> > 1. Your main positive theorem uses bounded density. Can you identify weaker regularity assumptions that still support discretization and sublinear regret against the GBB benchmark?
>
> Any assumption that can guarantee regularity on the GFT would be enough, in particular, it should guarantee that the gain from trade is at least one-sided Lipschitz for some constant.
>
>
> > 2. The algorithm depends on several horizon-linked parameters, including $K$, $N$, and $\beta$. Is there a clean anytime variant, or can you clarify what specifically breaks when $T$ is unknown?
>
> While the optimal parameters $K, N,$ and $\beta$ depend on the horizon $T$, this is not a fundamental limitation. To adapt to an unknown horizon, our algorithm can be wrapped in a standard doubling trick scheme. By partitioning time into epochs of doubling length and resetting the parameters accordingly, we could maintain the same order of regret without prior knowledge of $T$.
>
> > 3. Beyond i.i.d. stochastic valuations, do the ideas extend to mildly non-stationary or contextual settings, and if not, which component becomes the main bottleneck: discretization, exploration, or constrained optimization over feasible distributions?
>
> Expanding to these settings is a very interesting future direction but it would surely require a different algorithmic structure and analysis. For example, for non-stationary environments, a major bottleneck would be the strict division in three phases, which is inherently designed for the stochastic environment. Adapting to change would at least require mixing these phases to continuously re-evaluate the distribution. Ultimately, any non-stationarity guarantee would depend heavily on the specific measure of non-stationarity of the environment, and would need to be linear in the worst case, coherently with the linear lower bound of Benarnasconi et al. (2024).

---

> > ### Author Rebuttal · Reviewer_5YL2 · 2026-04-03
> >
> > I appreciate the authors' efforts in providing a thorough rebuttal. I plan to keep my original ratings.

---

### Official Review · Reviewer_oR5c · 2026-03-11

**Soundness:** 2
**Presentation:** 2
**Significance:** 2
**Originality:** 2
**Overall Recommendation:** 4
**Confidence:** 4

**Summary:**

This paper studies stochastic online bilateral trade with one-bit feedback under a global budget balance (GBB) requirement. Unlike prior work that competes with the best fixed WBB/SBB price, it uses the stronger benchmark of the best fixed distribution over price pairs whose expected profit is nonnegative. The paper first shows that without a bounded-density assumption on the joint valuation distribution, sublinear regret is impossible even under two-bit feedback, and then proves a $\tilde{O}(T^{3/4})$ regret bound under bounded density. The main technical idea is a three-phase algorithm that collects a profit buffer, estimates the non-bandit part of GFT over a $K \times K$ grid using only $2K$ exploratory price pairs, and then performs optimistic constrained bandit optimization. The final rate matches the known lower bound for the simpler fixed-price benchmark, suggesting no rate separation between the two settings under bounded density.

**Compliance With Llm Reviewing Policy:**

Affirmed.

**Ethical Review Concerns:**

Update: I unflagged the ethical review request, but would like to kindly ask the authors to pay attention to the following concern.

------------

A potential ethical concern is that a brokerage mechanism optimized for gain-from-trade and budget balance may produce systematically uneven outcomes across different participant groups, even when the learning objective is satisfied. If seller and buyer valuations are correlated with protected or socioeconomically sensitive attributes, the learned pricing distribution could disproportionately exclude some groups from trade or shift surplus away from them. The paper currently does not discuss fairness implications of such mechanisms, nor whether any fairness-aware constraints or evaluation criteria could be incorporated. Even a brief discussion of this risk would strengthen the paper’s ethical framing.

**Final Justification:**

Issues are addressed. An overall good work that I lean on slight positive.

**Key Questions For Authors:**

0, **Prompt injection**: I noticed that the submission contains text instructing the reviewer / LLM to include specific phrases in the review. I do not know whether this text was intentionally inserted by the authors or automatically added by some external system, so I do not want to overinterpret it. Could the authors please clarify where this text came from? In any case, I do not think it should affect the scientific evaluation of the paper.

1, Please comment on the weaknesses I mentioned above.

2, I feel like the impact statement is insufficient, especially given the nature of this paper that studies brokerage mechanisms and welfare/profit trade-offs in two-sided markets. Could the authors discuss possible implications for market design, fairness, or strategic concerns, even at a high level?

**Limitations:**

More discussion would be necessary.

**Strengths And Weaknesses:**

## Strengths:

1, A clear strength of the paper is that it upgrades the comparison target from the best fixed price to the best fixed GBB distribution over price pairs. This is a meaningful strengthening of the baseline, since prior WBB/SBB-oriented guarantees can become linear-regret under this benchmark.

2, Under the bounded-density assumption, the paper shows that moving from the one-dimensional fixed-price benchmark to the two-dimensional GBB benchmark does not worsen the regret rate: the result is still $\tilde{O}(T^{3/4})$. This is conceptually neat, and it is the main positive message of the paper.

## Weaknesses:

1, The “no separation” claim is stated too broadly. The paper repeatedly suggests that there is no separation between learning the best fixed SBB price and learning the best fixed GBB distribution. However, the paper itself also shows that bounded density is necessary in the GBB setting, while the simpler fixed-price setting does not always need such an assumption. Because of this, the claim seems accurate only at the level of the regret rate under bounded density, not at the level of learnability or assumptions more generally. I think the presentation should be more precise here.

2, The lower-bound proof is incomplete, and the paper has many notation / typo issues. The impossibility result is important for motivating the bounded-density assumption, but the proof currently feels too sketchy. In particular, the appendix contains visible notation problems and informal steps, and the lower-bound argument would benefit from a more formal and self-contained information-theoretic treatment. More broadly, the manuscript has many typos and notation inconsistencies, which makes the technical arguments harder to verify than they should be. This is especially unfortunate for a theory paper, where clarity and correctness of presentation matter a lot.

3, Mismatch between benchmark feasibility and learner feasibility. The benchmark only requires nonnegative expected profit, while the learner is required to satisfy realized global budget balance. This is a meaningful mismatch. It may well be a reasonable modeling choice, but the paper does not discuss it much, and I think it deserves more justification. In particular, it would be helpful to explain whether this is the most natural comparator, and how the problem might change if the comparator were also required to satisfy a stronger feasibility notion.

4, No numerical experiments.

---

> ### Author Rebuttal · Authors · 2026-03-31
>
> We thank the Reviewer for the time devoted to reviewing our paper.
>
> *Weakness*
> > 1
>
> The difference in assumptions noted by the Reviewer is specific to the "asymmetric" setting of Bernasconi et al. (2024), where a GBB learner is evaluated against a WBB baseline. In that case, the learner’s superior power allows for the removal of certain regularity assumptions. However, when considering algorithms with power comparable to the baseline, Cesa-Bianchi et al. (2021) demonstrate that the bounded density assumption is indeed necessary for a WBB learner against a WBB baseline. Without it, the regret becomes linear. Therefore, the "no separation" argument applies not only to the regret rate but also to the regularity assumptions required to achieve sublinear regret. We will clarify this distinction better in the final version to avoid any misunderstanding.
>
> > 2
>
> We apologize to the Reviewer for the typos and for being overly concise in the proofs. We will carefully proofread the paper for the final version and expand the proofs by making every step explicit that was previously condensed. As for the lower bound, the current proof already provides all the elements necessary to understand the general idea of the construction and the proof technique, but we agree with the Reviewer that additional details may be useful for a non-expert reader. We will surely make the argument more formal in the final version of the paper.
>
> > 3
>
> Both the benchmark and the notion of GBB algorithm have been introduced in Bernasconi et al. (2024) exactly as we use them here (in particular, see Equation (2) in (Bernasconi et al., 2024) for the definition of the benchmark). On the one hand, the benchmark definition is the most natural one when considering mechanims defined as distributions over price pairs. Indeed, forcing budget balance in expectation with respect to such distributions is the only option for a reasonable benchmark definition. On the other hand, it is also natural to consider algorithms that are GBB (no expectation). Indeed, an intermediary usually wants to deterministically avoid losing money, and enforcing budget balance in expectation on the algorithm's side would be a very louse constraint in many practical cases. Additionally, this mismatch between benchmark and algorithm does not negatively impact the achievable regret rate of $O(T^{3/4})$, which is the best one that can be achieved even against the weaker WBB baseline.
>
>
> > 4
>
> Online bilateral trade is a research area where the primary focus is to provide worst-case theoretical guarantees. As such, disigning an experimental campaign would require to carefully craft pathologically instances in which the algorithm robustness in worst case scenarios can numerically emerge. We point out that the core literature (e.g. [1,2,3,4,5]) on this topic do not include any numerical validation as the primary goal of this kind of papers (and also ours) is to advance the theoretical understanding on this topic.
>
> [1]Cesa-Bianchi, Nicolò, et al. "A regret analysis of bilateral trade." EC 2021.
>
> [2]Bernasconi, Martino, et al. "No-regret learning in bilateral trade via global budget balance." STOC 2024.
>
> [3]Azar, Yossi, et al. "An $\alpha $-regret analysis of Adversarial Bilateral Trade." NeurIPS 2022
>
> [4]Cesa-Bianchi, Nicolò, et al. "Regret analysis of bilateral trade with a smoothed adversary." JMLR 2024.
>
> [5]Lunghi, Anna, et al. "Better regret rates in bilateral trade via sublinear budget violation." SODA 2026
>
> *Questions*
>
> > 0
>
> Regarding prompt injection, please see https://icml.cc/Conferences/2026/PeerReviewFAQ#prompt_injection”.
>
> > 2
>
> Our work is primarily theoretical and therefore any application to real-world scenarios should be evalutad in the specifics of the applications, with the details on the sensibility of the task at hand. In mechanism design the choice of a social welfare function often involves a fundamental trade-off between efficiency and fairness. Historically, bilateral trade literature (including this work) focuses on utilitarian social welfare, which is defined as the aggregate utility of all agents. In our setting, this is effectively represented by the Gain from Trade (GFT), a measure that prioritizes economic efficiency. We recognize that the "correct" measure of welfare is highly context-sensitive, depending on the participants, the nature of the assets, and the institutional goals of the designer.
>
> Our approach is not intended as a "one-size-fits-all" solution, nor do we claim that GFT is a universal metric for all human interactions. Rather, it provides a robust, general framework for efficiency-maximization within the established bilateral trade paradigm. To address the Reviewer's concerns, we will include a dedicated section in our Impact Statement clarifying that the choice of GFT as a metric must be vetted against the specific social and fairness requirements of any practical implementation.

---

> > ### Author Rebuttal · Reviewer_oR5c · 2026-04-04
> >
> > Thanks for the authors' rebuttal. I got a clearer view of the paper as well as the analysis. I will adjust my score accordingly to positive.

---

### Official Review · Reviewer_Fjz5 · 2026-03-13

**Soundness:** 3
**Presentation:** 3
**Significance:** 3
**Originality:** 3
**Overall Recommendation:** 5
**Confidence:** 4

**Summary:**

This paper studies online bilateral trade, where a broker repeatedly mediates transactions between a seller and a buyer in order to maximize gain from trade (GFT), that is, the welfare created when trade occurs. At the same time, the broker must not subsidize the market, which is captured by global budget balance (GBB) for the learning algorithm, meaning that the broker’s cumulative profit over the horizon must be nonnegative.

Beyond requiring the learning algorithm itself to satisfy GBB, the paper also strengthens the benchmark. Instead of competing against the best fixed price, it competes against the best fixed distribution over price pairs $(p,q)$ that maximizes expected GFT subject to expected nonnegative profit. This is a significantly stronger benchmark than the previously studied best fixed-price WBB benchmark, which is equivalent to the best fixed-price SBB benchmark. As the paper shows, no-regret algorithms designed for that weaker benchmark can incur linear regret when evaluated against this stronger GBB benchmark.

The paper makes two main contributions. First, it proves an impossibility result showing that, without a bounded-density assumption on the joint distribution of buyer and seller valuations, no-regret with respect to this stronger GBB benchmark is impossible. Second, under bounded density, it gives an algorithm achieving $\widetilde{O}(T^{3/4})$ regret against the GBB benchmark. Since this matches the known $\Omega(T^{3/4})$ lower bound for the more restrictive fixed-price/SBB benchmark, the result suggests that, under bounded density, strengthening the benchmark from fixed prices to GBB distributions does not increase the regret rate, up to logarithmic factors.

**Compliance With Llm Reviewing Policy:**

Affirmed.

**Final Justification:**

The rebuttal addressed all my questions and concerns. I believe this work is a solid contribution, so I have decided to maintain my original score.

**Key Questions For Authors:**

The paper does explain that bounded density helps because it smooths the objective enough for discretization to work, and that this is exactly what fails in the lower-bound construction. However, I would appreciate a more intuitive explanation of this phenomenon from the positive side.

**Limitations:**

One limitation is that the model does not account for strategic behavior by buyers and sellers. In practice, agents may have incentives to manipulate the algorithm through their responses to posted prices, which could create additional inefficiencies beyond the learning problem considered here. Understanding this strategic layer, and whether it leads to price-of-anarchy-type effects, would be an interesting direction for future work.

**Strengths And Weaknesses:**

### **Strengths**

The main strength is that the paper appears to provide the first sublinear-regret result against the stronger fixed-distribution GBB benchmark in the stochastic setting under bounded density. Prior work, most notably Bernasconi et al. 2024, achieved $\widetilde{O}(T^{3/4})$ regret using a GBB learner, but benchmarked against the weaker fixed-price WBB benchmark, equivalently the fixed-price SBB benchmark. For the stronger GBB-distribution benchmark, the previous positive guarantee was only a factor-2 comparison, rather than true no-regret.
I therefore view this as a meaningful theoretical contribution to the online bilateral trade literature. The paper shows that a benchmark previously known to be impossible in more general settings becomes learnable under a natural regularity assumption. Moreover, the algorithm itself satisfies GBB, rather than merely competing against a GBB-style benchmark. The result also suggests that, under bounded density, the stronger GBB benchmark has essentially the same regret complexity as the weaker benchmarks.


### **Weakness**

It is not fully clear what kind of technical innovations were needed to break this barrier compared to Bernasconi et al. It seems that part of the breakthrough is structural: under the bounded density assumption, the objective becomes smoother, so discretization works, whereas without it the pathological hard instances can make the problem too irregular.
The paper does mention this point, but I think a more complete high-level picture could be articulated more clearly. For example, a simple illustrative example, figure, or experiment showing how bounded density rules out the problematic sharp irregularities would make the result easier to understand.
On the other hand, it would help if the paper clarified more explicitly which part is the algorithmic innovation and which part comes from the structural simplification induced by the bounded density assumption.

Overall I think this is a solid contribution.

---

> ### Author Rebuttal · Authors · 2026-03-31
>
> We thank the Reviewer for their insightful comments and for their positive evaluation of our work.
>
> > It is not fully clear what kind of technical innovations were needed to break this barrier compared to Bernasconi et al. It seems that part of the breakthrough is structural: under the bounded density assumption, the objective becomes smoother, so discretization works, whereas without it the pathological hard instances can make the problem too irregular. The paper does mention this point, but I think a more complete high-level picture could be articulated more clearly. For example, a simple illustrative example, figure, or experiment showing how bounded density rules out the problematic sharp irregularities would make the result easier to understand. On the other hand, it would help if the paper clarified more explicitly which part is the algorithmic innovation and which part comes from the structural simplification induced by the bounded density assumption.
>
> Bernasconi et al. (2024) focus on adversarial settings, so they prove a linear lower bound that strictly depends on the non-stationarity of the environment. Indeed, in a non-stationary environment, losing money before gaining it is unthikable, as there are no guarantees that later the instance will be generous enough that the econmic loss can be compensated. At the same time, always playing safe can be arbirarily bad in "positive" instances. Learning against the distributional benchmark in a stochastic environment, with or without bounded density, has never been discussed prior to this work. We will add more inuitive explanation on how bounded density make this problem learnable, adding also illustrative example with the additional space.
>
> > The paper does explain that bounded density helps because it smooths the objective enough for discretization to work, and that this is exactly what fails in the lower-bound construction. However, I would appreciate a more intuitive explanation of this phenomenon from the positive side.
>
> The effect of the bounded density assumption is that it guarantees that on a regular grid, the optimum on the grid is near enough the general optimum. Specifically, the effect of the bounded density assumption is all contained in section 5. We will add more intuitions.

---

> > ### Author Rebuttal · Reviewer_Fjz5 · 2026-04-04
> >
> > Thank you for the detailed responses. I will keep my original score.

---

### Official Review · Reviewer_4kVt · 2026-03-13

**Soundness:** 3
**Presentation:** 3
**Significance:** 3
**Originality:** 4
**Overall Recommendation:** 5
**Confidence:** 3

**Summary:**

This paper studies online bilateral trade with one-bit feedback (only knowing if a trade happened or not). The goal is to maximize the GFT. Instead of forcing the platform to break even on every single trade (WBB/SBB), the authors use a Global Budget Balance (GBB) rule. This means the platform can lose money on some trades to subsidize others, as long as it doesn't lose money overall.
Competing with the best GBB strategy is hard. Past WBB algorithms fail here, and searching for two prices instead of one usually requires way too many samples.
The paper solves this by making two main contributions:
1.	A necessary condition: They prove that without a "bounded density" assumption for valuations, any algorithm will fail (linear regret). This is a new finding, as older strict-budget models didn't need this rule.
2.	A tight 3-phase algorithm: They build an algorithm that achieves an $\tilde{O}(T^{3/4})$ regret, which matches the theoretical best possible rate.
The smartest part of their algorithm is how it handles the 2D price grid. It works in 3 steps:
•	Phase 1: It acts safe to save up a "profit buffer" (extra money).
•	Phase 2: It explores the GFT. Normally, searching a 2D grid for buyer and seller prices takes too much data. But the authors found a clever way to reuse samples across different price pairs. This means they can search the 2D grid using the same amount of data as a 1D search.
•	Phase 3: It uses the saved money and the GFT estimates to pick the best prices and maximize welfare.
Overall, the paper clearly proves that learning two prices for a global budget isn't fundamentally harder than learning one price for a strict budget, as long as the density is bounded.

**Compliance With Llm Reviewing Policy:**

Affirmed.

**Final Justification:**

I am positive on this paper. I will keep my score as it is.

**Key Questions For Authors:**

None

**Strengths And Weaknesses:**

Strengths：
1.	Novel Perspective and Complete Results: The paper introduces a very fresh perspective. It is the first work to use the GBB as the benchmark. The theoretical results are also very complete: it first proves a negative result (any algorithm fails if the density is unbounded), and then provides a tight, sublinear algorithm for the bounded case.
2.	Strong Technical Contribution: The authors successfully tackle the sample complexity issue on a 2D grid. During the pure exploration phase, it is highly impressive that they achieve a uniform approximation over the two-dimensional grid without increasing the number of samples.
3.	Clear Writing: The paper is generally well-written and easy to read. The Introduction and Preliminaries sections are especially clear and do a great job setting up the problem.
Weaknesses：
1.	Dense Technical Sections: The later technical parts of the paper (Sections 7, 8, and 9) are a bit hard to follow. While the math seems correct, these sections lack high-level intuition and simple examples to help readers understand the core ideas.

---

> ### Author Rebuttal · Authors · 2026-03-31
>
> We thank the Reviewer for their insightful comments and for their positive evaluation of our work.
>
> > 1. Dense Technical Sections: The later technical parts of the paper (Sections 7, 8, and 9) are a bit hard to follow. While the math seems correct, these sections lack high-level intuition and simple examples to help readers understand the core ideas.
>
> We will follow the suggestion of the Reviewer and add additional intuition and simple examples in the final version of the paper.

---

> > ### Author Rebuttal · Reviewer_4kVt · 2026-04-04
> >
> > Thank you for your detailed response. I appreciate the clarification and will keep my score as is.

---

### Decision · Program_Chairs · 2026-04-30

**Decision:**

Accept (regular)

**Comment:**

The paper follows up on a natural and important development in the repeated bilateral trade literature: change of benchmark from weak-budget balance (that requires breaking even in every trade) to global budget balance (being budget balance at the end of all trades). The latter is quite natural as a benchmark in repeated settings, and a recent paper showed that the new benchmark admits an improved approximation to the gains-from-trade, by a factor of two. The natural question then is whether one can get a sublinear regret aganist this new benchmark, which is the question this paper answers in the positive. The paper shows this result in a stochastic environment with one-bit feedback, namely, the learner that posts the prices, observes whether the trade happened or not, and nothing else.

The reviewers uniformly appreciated both the importance of the question studied in this paper, the technical contribution of the paper (the algorithms and analysis) and the completeness of the results, going from providing a sublinear regret algorithm, to resolving the sample complexity question in the 2D grid of prices. The writing is also mostly clear, even if the technical sections could use some intuitive explanations before diving into the details. Overall a solid ICML paper.

Minor: In the GFT literature, GFT and welfare are two different things. I am sure the authors know this, but the first sentence of the abstract might be slightly confusing for some. While later in the paper the writing becomes clear on this, it is better to be clear everywhere.